# Phases of adjoint QCD$_3$ and dualities

**Jaume Gomis[1] Zohar Komargodski[2,3] and Nathan Seiberg[4]**

**1** Perimeter Institute for Theoretical Physics, Waterloo, Ontario, N2L 2Y5, Canada
**2** Department of Particle Physics and Astrophysics, Weizmann Institute of Science, Israel
**3** Simons Center for Geometry and Physics, Stony Brook University, Stony Brook, NY
**4** School of Natural Sciences, Institute for Advanced Study, Princeton, NJ 08540, USA

## Abstract

We study 2+1 dimensional gauge theories with a Chern-Simons term and a fermion in the adjoint representation. We apply general considerations of symmetries, anomalies, and renormalization group flows to determine the possible phases of the theory as a function of the gauge group, the Chern-Simons level $k$, and the fermion mass. We propose an inherently quantum mechanical phase of adjoint QCD with small enough $k$, where the infrared is described by a certain Topological Quantum Field Theory (TQFT). For a special choice of the mass, the theory has $\mathcal{N} = 1$ supersymmetry. There this TQFT is accompanied by a massless Majorana fermion – a Goldstino signaling spontaneous supersymmetry breaking. Our analysis leads us to conjecture a number of new infrared fermion-fermion dualities involving $SU$, $SO$, and $Sp$ gauge theories. It also leads us to suggest a phase diagram of $SO/Sp$ gauge theories with a fermion in the traceless symmetric/antisymmetric tensor representation of the gauge group.

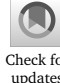
# 1 Introduction

Consider adjoint QCD in 2+1 dimensions, that is Yang-Mills theory with a Chern-Simons term at level $k$ and a Majorana fermion $\lambda$ in the adjoint representation of the gauge group $G$. We study the low energy dynamics of this theory as a function of the gauge group $G$, the Chern-Simons level $k$ and the mass $M_\lambda$ of the fermion. We uncover a rich structure of phase diagrams that suggest new fermion-fermion dualities and include phases that are inherently quantum mechanical (in the sense that they are invisible semiclassically).

The infrared behaviour of the theory crucially depends on the level $k$ of the Chern-Simons term. We denote by $G_k$ a Chern-Simons term with gauge group $G$ and level $k$. We will be mostly interested in the classical gauge groups $G = SU(N)$, $SO(N)$, $Sp(N)$ and follow the conventions of [1–5].[1] The Lagrangian of adjoint QCD contains a Chern-Simons term with coefficient $k_{bare}$ which must be properly quantized. Since the theory has a fermion, we henceforth consider the theory on spin manifolds, which requires that $k_{bare} \in \mathbb{Z}$. The level $k$ is related to $k_{bare}$ by

$$k = k_{bare} - \frac{h}{2} \tag{1.1}$$

with $h$ the dual Coxeter number of $G$, e.g.[2] The virtue of labeling the theory by $k$ is that time-reversal (or parity) acts by $k \to -k$ (alongside with reversing the sign of the fermion mass $M_\lambda \to -M_\lambda$). Without loss of generality we henceforth consider $k \geq 0$. Note that adjoint QCD for $k = 0$ and a vanishing mass ($M_\lambda = 0$) for the fermion is time-reversal invariant. This time-reversal invariant theory therefore exists in $SU(N)$ or $SO(N)$ only for $N$ even and in $Sp(N)$ only for $N$ odd.

Adjoint QCD has two semiclassically accessible phases for all $G$ and $k$. When the mass of the fermion is much larger than the scale set by the Yang-Mills coupling (i.e. $|M_\lambda| \gg g^2$) the fermion can be integrated out at one-loop. This shifts the coefficient of the Chern-Simons level [10–12]

$$k \to k + \text{sgn}(M_\lambda)\frac{h}{2}. \tag{1.2}$$

At low energies the Yang-Mills term becomes irrelevant and the infrared description is pure Chern-Simons TQFT with gauge group $G$ and level $k + h/2$ for large positive mass and level $k - h/2$ for large negative mass. We denote these Chern-Simons theories by $G_{k+h/2}$ and $G_{k-h/2}$ respectively. Our goal is to fill in the rest of the phase diagram.

For a special value of the bare mass

$$M_\lambda = m_{SUSY} \sim -kg^2, \tag{1.3}$$

adjoint QCD is $\mathcal{N} = 1$ supersymmetric.[3] At this point the fermion $\lambda$ is the gaugino in the $\mathcal{N} = 1$ vector multiplet. Moving away from this supersymmetric point in the phase diagram can be interpreted as turning on a soft supersymmetry-breaking mass $m_\lambda$ for the gaugino

$$M_\lambda = m_{SUSY} + m_\lambda. \tag{1.4}$$

In the quantum theory these masses are corrected and the theory may end up being massless even if the bare Lagrangian has a mass term. If $h/2$ is a half-integer so is $k$ and if $h/2$ is an integer so is $k$. In other words, we must take $k = \frac{h}{2} \bmod 1$, which in our cases implies

---

[1] Our notation is $Sp(1) = SU(2)$.

[2] Note that the dual Coxeter number of $SO(3)$ is one, while that of $SU(2)$ is two. Correspondingly, our notation is $SO(3)_k = SU(2)_{2k}/\mathbb{Z}_2$. More generally, we label the TQFT by the corresponding Chern-Simons gauge group and its level. A quotient as in this expression is interpreted from the 2d RCFT as an extension of the chiral algebra [6] and from the 3d Chern-Simons theory as a quotient of the gauge group [7]. More abstractly, it can be interpreted as gauging a one-form global symmetry of the TQFT [8,9]. This quotient is referred to in the condensed matter literature as "anyon condensation."

[3] $\mathcal{N} = 1$ supersymmetry in 2+1 dimensions entails 2 real supercharges.

| $G$ | $SU(N)$ | $SO(N)$ | $Sp(N)$ |
|---|---|---|---|
| $h$ | $N$ | $N-2$ | $N+1$ |

Table 1: Dual Coxeter number for classical groups.

| $G$ | $SU(N)$ | $SO(N)$ | $Sp(N)$ |
|---|---|---|---|
| $k$ | $\frac{N}{2}\bmod 1$ | $\frac{N}{2}\bmod 1$ | $\frac{N+1}{2}\bmod 1$ |

Table 2: Quantization of Chern-Simons levels in adjoint QCD.

Let us consider the infrared dynamics of the supersymmetric theory; i.e. set $m_\lambda = 0$. For large $k$ all the propagating degrees of freedom have mass of order $kg^2 \gg g^2$ and the semiclassical analysis is again reliable. First, we integrate out the gauginos thus shifting the Chern-Simons coefficient [13, 14]

$$k_{IR} = k - \frac{h}{2}\,. \tag{1.5}$$

Second, the gauge fields are heavy and can be integrated out (except for some global modes) leading at low energies to a topological $G_{k-\frac{h}{2}}$ theory. Witten argued that the large $k$ infrared description $G_{k-\frac{h}{2}}$ remains valid in the entire domain of $k$ where supersymmetry is unbroken [14], that is for

$$k \geq \frac{h}{2}\,. \tag{1.6}$$

Thus, the low energy description at the supersymmetric point is given by $G_{k-\frac{h}{2}}$ Chern-Simons theory. This infrared theory coincides with the asymptotic phase at large negative mass.[4]

This suggests a natural scenario for the phase diagram of adjoint QCD for $k \geq \frac{h}{2}$ depicted in fig. 1. The system has two asymptotic phases with topological order $G_{k\pm\frac{h}{2}}$. The supersymmetric point $m_\lambda = 0$ is in the $G_{k-\frac{h}{2}}$ phase. The two phases are separated by a transition at some positive value of $m_\lambda$. At that point we can think of the fermion as being effectively massless. We do not actually know whether this transition is first order or second order. However, for very large $k$ we can think about the model perturbatively, starting from the CFT of $\dim(G)$ free fermions. The anomalous dimensions of the $G$-invariant local operators are only weakly corrected compared to the free model and the transition is second order. Therefore, it is natural to expect that for eq. (1.6) the transition between the two topological theories $G_{k+h/2}$ and $G_{k-h/2}$ is second order.

Let us now turn to the interesting region $k < \frac{h}{2}$. We know from our discussion above that there are two asymptotic regions of large mass described by the TQFTs $G_{k\pm h/2}$. The infrared dynamics for small mass $m_\lambda$ is strongly coupled and requires a more sophisticated analysis. We will see that the physics at small $m_\lambda$ can lead to novelties in comparison to the phase diagram for $k \geq h/2$.

At the supersymmetric point $m_\lambda = 0$ supersymmetry is expected to be spontaneously broken for $k < h/2$ [14]. This implies that at $m_\lambda = 0$ there is a massless Majorana fermion (i.e. the Goldstino particle). An interesting question is whether the infrared theory contains a TQFT in addition to the Majorana Goldstino particle. We will see that general considerations including

---

[4]Note that for $k = \frac{h}{2}$, $k_{IR} = 0$ and hence the infrared theory is rather simple. For simply connected gauge groups it is completely trivial. But for non-simply connected gauge groups, like $SO(N)_0$, it has a zero-form magnetic symmetry, which can be spontaneously broken, leading to several vacua in the infinite volume system.

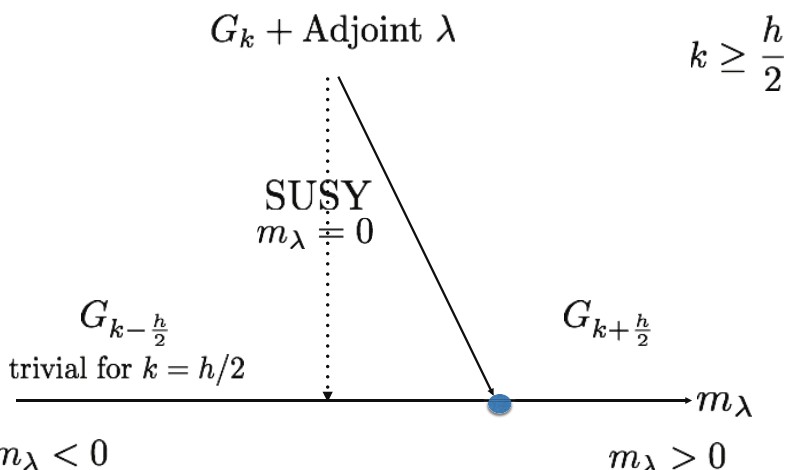

Figure 1: The phase diagram of $G_k$ with an adjoint fermion for $k \geq \frac{h}{2}$. Note that the physics at the supersymmetric point is smooth – there is no transition there. For $k = \frac{h}{2}$ the negative mass phase is trivial and there is a transition at some positive value of $m_\lambda$. Here, and in all our later figures, the vertical line denotes renormalization group flow from the UV at the top of the diagram to the IR at the bottom of the diagram. The solid diagonal line represents a flow to an IR fixed point. The physics across this line is not smooth. The theory is supersymmetric along the dotted vertical line. Unlike the solid line, here the physics is smooth as this line is crossed.

symmetries and 't Hooft anomaly matching imply that this is indeed the case.[5] We will also identify the TQFT accompanying the Goldstino.

A first guess for the phase diagram of adjoint QCD for $k < h/2$ is that there are two phases, as in fig. 1, except that at the point $m_\lambda = 0$ there is also a massless Goldstino reflecting the fact that supersymmetry is spontaneously broken. In fact, we argue below that for $G = SU(N)$, $SO(N)$, and $Sp(N)$ this is the correct scenario, but only for a special value of $k = \frac{h}{2} - 1$.[6] This is depicted in fig. 2.[7] Note that the low energy theory around the point of the massless Goldstino includes a decoupled TQFT $G_{-1}$. There is a phase transition to the right of the supersymmetric point $m_\lambda = 0$, which we denote by a bullet point. We will not be able to determine whether this transition is first or second order. We find that this phase transition admits an alternative description in terms of another gauge theory coupled to a fermion in a representation of the gauge group that depends on the choice of $G$. This suggests new fermion-fermion dualities in nonsupersymmetric theories (see below). For other recent nonsupersymmetric dualities see [2–4, 15–25].

We are going to argue that the scenario in fig. 2 cannot be right for all lower values of $k$, i.e. all $k < \frac{h}{2} - 1$. To see that, consider the special case of $k = 0$. We know that for large $|m_\lambda|$ we have the two asymptotic topological phases $G_{\pm \frac{h}{2}}$ exchanged by the action of time-reversal. (Time-reversal changes the sign of $M_\lambda$ and indeed exchanges the two asymptotic phases.) Also, since supersymmetry is spontaneously broken, we expect a massless Goldstino at the supersymmetric point $M_\lambda = m_\lambda = 0$. If the system has only the two topological phases $G_{\pm h/2}$, the supersymmetric point must be at the transition point. So one might be tempted to consider a phase diagram with only two phases, one for $m_\lambda$ positive and the other for $m_\lambda$

---

[5]With the exception of adjoint QCD with gauge group $SO(N)$ and a Chern-Simons term at level $k = h/2 - 1$ where the infrared description is just a Goldstino.

[6]In our discussion below we will mostly exclude the case $SO(4)_0 = (SU(2)_0 \times SU(2)_0)/\mathbb{Z}_2$, where supersymmetry is broken with two massless Goldstinos, one for each $SU(2)$ factor.

[7]We have not analyzed whether this picture is valid also for other gauge groups.

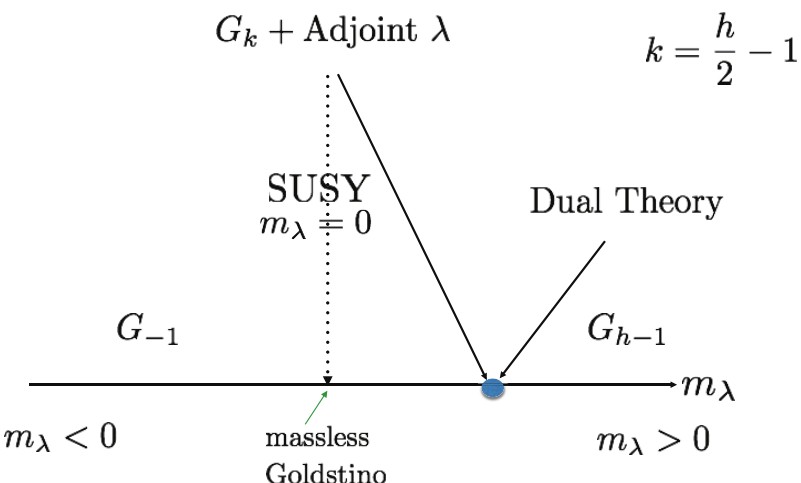

Figure 2: The proposed phase diagram for $G = SU(N)$, $SO(N)$, and $Sp(N)$ with $k = \frac{h}{2} - 1$. For low values of $N$ the situations can be different. For example, for $SU(2)_0$ the transition point occurs at the supersymmetric point $m_\lambda = 0$. And other than the massless Goldstino there, the TQFT does not change because of the duality $SU(2)_{-1} \longleftrightarrow SU(2)_1$ (as spin-TQFT). Below we will discuss another dual theory that flows to the same transition. It is denoted in the figure as "Dual Theory." The general structure of the figure is as explained in the caption of fig. 1, except that here supersymmetry is broken for $m_\lambda = 0$ and we also suggest a dual theory of the transition point.

negative. At the supersymmetric point the infrared theory must be time-reversal invariant. This would be a consistent scenario if the two topological phases are level/rank dual to each other, $G_{\frac{h}{2}} \longleftrightarrow G_{-\frac{h}{2}}$,[8] that is if the topological phase $G_{\frac{h}{2}}$ is time-reversal invariant.[9] We could then have a massless Goldstino at $m_\lambda = 0$ together with the decoupled TQFT $G_{\frac{h}{2}}$. However, this condition that $G_{\frac{h}{2}}$ is dual to $G_{-\frac{h}{2}}$ is not obeyed for generic $G$ and thus invalidates this simplistic scenario. Another possibility is that the theory at that supersymmetric point spontaneously breaks its time-reversal symmetry such that the transition at $m_\lambda = 0$ is first order. This possibility can be ruled out using the argument of [26]. Alternatively, the transition at that point could be second order and that would mean that the low energy theory includes more degrees of freedom. Such additional degrees of freedom would also need to match various anomalies of the ultraviolet adjoint QCD theory. We cannot exclude this possibility. But we will argue for another, more likely, and simpler option.

Let us now present our scenario for $k < \frac{h}{2} - 1$: a new intermediate quantum mechanical phase opens up between the two asymptotic phases. The details of the scenario are summarized in fig. 2. This picture is motivated by the analysis of QCD$_3$ in [5], where it was suggested that for small fundamental quark masses the system can have a new purely quantum phase, which cannot be understood semiclassically. It is also motivated by studying the effective field theory on domain walls and interfaces along the lines of [27–30], as well as by the holographic realization [31,32] of adjoint QCD for $m_\lambda = 0$ from which the intermediate infrared TQFT can be extracted by analyzing the effective low energy theory on branes wrapping a noncontractible cycle threaded by flux. A detailed analysis of these domain walls and interfaces will appear in [30].

General considerations imply that the TQFT in the intermediate, new phase cannot be

---

[8]We use the symbol $A \longleftrightarrow B$ to denote that theories $A$ and $B$ are dual.

[9]This is true for $G = SU(2)$ where $SU(2)_1 \longleftrightarrow SU(2)_{-1}$, but this case has $k = \frac{h}{2} - 1 = 0$ and here we discuss $k < \frac{h}{2} - 1$. See a comment in fig. 2.

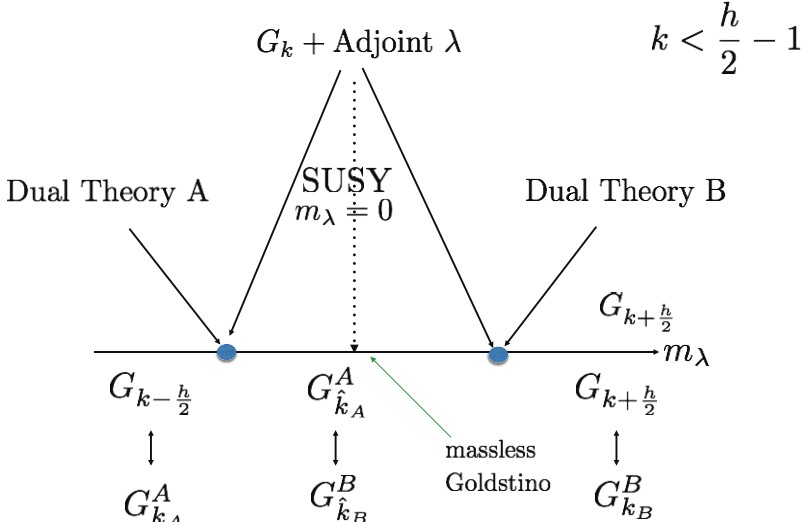

Figure 3: The proposed phase diagram for $k < \frac{h}{2} - 1$. The system has three phases. Two of them at large $|m_\lambda|$ are visible semiclassically and the middle phase at small $|m_\lambda|$ is purely quantum. The supersymmetric point $m_\lambda = 0$ is in the interior of this middle phase. At that point there is a massless Goldstino. Even when the middle phase is gapped it includes some TQFT, which we will determine. We will also present two conjectured dual gauge theories, denoted by "Dual Theory A" and "Dual Theory B" that describe the two transition points. Again, the general structure of this diagram is as explained in the caption of fig. 1, except that here we have two transition points and two dual theories.

| $SU(N)_k$ | $Sp(N)_k$ | $SO(2N)_{2k}$ | $SO(2N)_{2k+1}$ | $SO(2N+1)_k$ |
|-----------|-----------|---------------|-----------------|--------------|
| $\mathbb{Z}_N$ | $\mathbb{Z}_2$ | $\mathbb{Z}_2$ | 1 | 1 |

Table 3: One-form global symmetry of adjoint QCD.

trivial. Adjoint QCD has a one-form global symmetry (see table 3) acting on its line operators [8,9]. This symmetry cannot act trivially in the infrared (i.e. it cannot be that all the lines would be confined) because it has an 't Hooft anomaly for generic $k$. Therefore, this symmetry must also be realized in the infrared. But since a single Goldstino clearly does not match the one-form global symmetry of adjoint QCD, this rules out the scenario with just a Goldstino. This scenario can also be ruled out in the $T$-reversal-invariant cases $k = 0$. In these cases adjoint QCD has a $T$-reversal anomaly, characterized by an integer modulo 16, which is often denoted by $\nu$ [33–41]. (For early work on mixed $T$-gravitational anomalies, see e.g. [42].) The massless Goldstino does not saturate that anomaly.[10] Below we will discuss how the proposed TQFT in the infrared does saturate that anomaly.

We suggest that adjoint QCD for $k < \frac{h}{2} - 1$ has a new intermediate phase in between the asymptotic phases described by $G_{k\pm h/2}$. The intermediate phase is another gapped phase except at the supersymmetric point $m_\lambda = 0$ where there is a massless Goldstino. The TQFT describing the gapped phase differs from the asymptotic TQFTs $G_{k\pm h/2}$. Below we identify the TQFT that governs the intermediate phase for $G = SU(N)$, $SO(N)$, $Sp(N)$ gauge theory with

---

[10]We thank E. Witten for raising this point.

a fermion in the adjoint representation[11]

$$U\left(\frac{N}{2} - k\right)_{\frac{N}{2}+k,N} \qquad \text{for} \quad SU(N)_k \quad \text{with} \quad k < \frac{N}{2} - 1,$$

$$SO\left(\frac{N-2}{2} - k\right)_{\frac{N-2}{2}+k} \qquad \text{for} \quad SO(N)_k \quad \text{with} \quad k < \frac{N}{2} - 2, \tag{1.9}$$

$$Sp\left(\frac{N+1}{2} - k\right)_{\frac{N+1}{2}+k} \qquad \text{for} \quad Sp(N)_k \quad \text{with} \quad k < \frac{N-1}{2}.$$

Note that the intermediate TQFTs for $SU(N)$ adjoint QCD coincide with the domain wall theories of 3+1 dimensional $\mathcal{N} = 1$ $SU(N)$ super-Yang-Mills in [27]. This will be important in [30].

The phase diagram has two phase transitions connecting the intermediate phase to the asymptotic phases. We find that the phase transitions can be described by a different gauge theory with a fermion in a representation of the gauge group that depends on the choice of $G$. We denote these dual theories in fig. 3 by "Dual Theory A" and "Dual Theory B." Specifically, we propose the new fermion-fermion dualities:[12]

$SU(N)$ and $k < \frac{N}{2}$:

$$SU(N)_k + \text{adjoint } \lambda \longleftrightarrow U\left(\frac{N}{2} + k\right)_{-\frac{3}{4}N+\frac{k}{2},-N} + \text{adjoint } \tilde{\lambda},$$

$$SU(N)_k + \text{adjoint } \lambda \longleftrightarrow U\left(\frac{N}{2} - k\right)_{\frac{3}{4}N+\frac{k}{2},N} + \text{adjoint } \hat{\lambda}. \tag{1.10}$$

$SO(N)$ and $k < \frac{N-2}{2}$:

$$SO(N)_k + \text{adjoint } \lambda \longleftrightarrow SO\left(\frac{N-2}{2} + k\right)_{-\frac{3N}{4}+\frac{k}{2}+\frac{1}{2}} + \text{symmetric } \tilde{S},$$

$$SO(N)_k + \text{adjoint } \lambda \longleftrightarrow SO\left(\frac{N-2}{2} - k\right)_{\frac{3N}{4}+\frac{k}{2}-\frac{1}{2}} + \text{symmetric } \hat{S}. \tag{1.11}$$

$Sp(N)$ and $k < \frac{N+1}{2}$

$$Sp(N)_k + \text{adjoint } \lambda \longleftrightarrow Sp\left(\frac{N+1}{2} + k\right)_{-\frac{3N}{4}+\frac{k}{2}-\frac{1}{4}} + \text{antisymmetric } \tilde{A},$$

$$Sp(N)_k + \text{adjoint } \lambda \longleftrightarrow Sp\left(\frac{N+1}{2} - k\right)_{\frac{3N}{4}+\frac{k}{2}+\frac{1}{4}} + \text{antisymmetric } \hat{A}. \tag{1.12}$$

Both here and below when we discuss symmetric representations of $SO(N)$ and antisymmetric representations of $Sp(N)$ we mean the irreducible symmetric-traceless and antisymmetric-traceless representations. The dualities in the second line of eqs. (1.10) to (1.12) trivialize

---

[11]We follow the standard notation

$$U(M)_{P,Q} = \frac{SU(M)_P \times U(1)_{MQ}}{\mathbb{Z}_M},$$

$$U(M)_P \equiv U(M)_{P,P}. \tag{1.7}$$

$U(M)_{P,Q}$ is described by the Chern-Simons Lagrangian

$$\frac{P}{4\pi} \text{Tr}(a \wedge da + \frac{2}{3} a \wedge a \wedge a) + \frac{Q-P}{4\pi M}(\text{Tr}\, a) \wedge (d\, \text{Tr}\, a), \tag{1.8}$$

with $a$ a $U(M)$ gauge field. This expression makes it clear that it is well defined for integer $P, Q$ with $P = Q$ mod $M$ and that the theory is a spin TQFT when $P + \frac{Q-P}{M}$ is odd (see e.g. [1,3]).

[12]As in other familiar examples, including most recently [5], the notion of this duality is valid only near the transition point. We say that $G_k + \text{adjoint } \lambda$ is dual to Theory A around one transition and it is dual to Theory B around the other transition, but we do not say that Theory A is dual to Theory B.

when $k$ reaches the upper limit, i.e. when $k = h/2 - 1$. It should also be pointed out that the dualities in the second line of eqs. (1.10) to (1.12) can be viewed as "analytic continuation" to negative $k$ combined with orientation reversal of the dualities in the first line.

A way to understand fig. 3 is as follows. We start with large and negative $m_\lambda$ and find $G_{k-\frac{h}{2}}$. Then we use level/rank duality to express this theory as $G^A_{k_A}$ with some gauge group $G^A$ and some level $k_A$. The Dual Theory A has gauge group $G^A$ with some level and some matter fields. Specifically, this theory can be found in eqs. (1.10) to (1.12). Then we move to the right in fig. 3 and cross a phase transition. Here the matter fields in theory A change the sign of their mass and $k_A$ changes to $\hat{k}_A$. We end up in the intermediate phase with the TQFT $G^A_{\hat{k}_A}$ with some level $\hat{k}_A$. In our examples, these are the TQFTs in eq. (1.13). We then repeat these steps for large and positive $m_\lambda$, where again we use level/rank duality and then a transition in Dual Theory B to find the TQFT in the intermediate phase $G^B_{\hat{k}_B}$. Our proposal for the intermediate phase and the dual theories eqs. (1.10) to (1.12) was motivated by requiring that the two descriptions of the TQFT in the intermediate phase are dual to each other, i.e. $G^A_{\hat{k}_A} \longleftrightarrow G^B_{\hat{k}_B}$.

Below we will discuss this process in a lot of detail in our three examples $SU(N)$, $SO(N)$, and $Sp(N)$.

As with most dualities, these are merely conjectures. In fact, as we emphasized above, our entire proposed phase diagram fig. 3 is conjectural. A common check of dualities is the matching of their global symmetries and their 't Hooft anomalies. A simple argument shows that many of these tests are automatically satisfied in our proposal. We have just mentioned that our scenario was designed such that the two descriptions of the intermediate phase are dual to each other, $G^A_{\hat{k}_A} \longleftrightarrow G^B_{\hat{k}_B}$. Therefore, throughout our phase diagram we used either level/rank duality or a phase transition in a weakly coupled theory. This guarantees that all the symmetries that are preserved through this process and their 't Hooft anomalies must match. These include all the zero-form and the one-form global symmetries [8,9].

A notable exception to this statement is a symmetry that is violated in our process of changing $m_\lambda$. Consider the special theories with $k = 0$. For $m_\lambda = 0$ they are time-reversal invariant. This symmetry is not present as we vary the mass and get to the intermediate phase and therefore there is no guarantee that it is present there. As a nontrivial check, for $k = 0$ the TQFT in the intermediate phase eq. (1.13) is $T$-invariant

$$
\begin{aligned}
U\left(\frac{N}{2}\right)_{\frac{N}{2},N} &\longleftrightarrow U\left(\frac{N}{2}\right)_{-\frac{N}{2},-N} , \\
SO\left(\frac{N-2}{2}\right)_{\frac{N-2}{2}} &\longleftrightarrow SO\left(\frac{N-2}{2}\right)_{-\frac{N-2}{2}} , \\
Sp\left(\frac{N+1}{2}\right)_{\frac{N+1}{2}} &\longleftrightarrow Sp\left(\frac{N+1}{2}\right)_{-\frac{N+1}{2}} ,
\end{aligned}
\tag{1.13}
$$

where we used results in [3,4,43].

These $T$-reversal-invariant cases also lead to additional consistency checks. As we mentioned above, it is known that $T$-reversal-invariant $2+1$ dimensional spin theories are subject to an anomaly, which is characterized by an integer $\nu$ modulo 16. This integer must match between adjoint QCD and our proposed infrared theory, which includes the Goldstino and the TQFTs eq. (1.13). In order to do that we will need the value of $\nu$ for these TQFTs [44–47] (special cases had been found earlier)

$$
\begin{aligned}
\nu\left(U(n)_{n,2n}\right) &= \pm 2 \mod 16 , \\
\nu\left(SO(n)_n\right) &= \pm n \mod 16 , \\
\nu\left(Sp(n)_n\right) &= \pm 2n \mod 16 .
\end{aligned}
\tag{1.14}
$$

So far we discussed the fermion-fermion dualities eqs. (1.11) and (1.12) in the range of small $k$. It is natural to examine them also for larger $k$, which after redefining $k$, are written for $SO(N)_k$ with $k < \frac{N+2}{2}$ as

$$SO(N)_k + \text{symmetric } S \longleftrightarrow SO\left(\frac{N+2}{2} + k\right)_{-\frac{3N}{4} + \frac{k}{2} - \frac{1}{2}} + \text{adjoint } \tilde{\lambda}$$

$$SO(N)_k + \text{symmetric } S \longleftrightarrow SO\left(\frac{N+2}{2} - k\right)_{\frac{3N}{4} + \frac{k}{2} + \frac{1}{2}} + \text{adjoint } \hat{\lambda}. \quad (1.15)$$

and for $Sp(N)_k$ with $k < \frac{N-1}{2}$ as

$$Sp(N)_k + \text{antisymmetric } A \longleftrightarrow Sp\left(\frac{N-1}{2} + k\right)_{-\frac{3N}{4} + \frac{k}{2} + \frac{1}{4}} + \text{adjoint } \tilde{\lambda},$$

$$Sp(N)_k + \text{antisymmetric } A \longleftrightarrow Sp\left(\frac{N-1}{2} - k\right)_{\frac{3N}{4} + \frac{k}{2} - \frac{1}{4}} + \text{adjoint } \hat{\lambda}. \quad (1.16)$$

The duality in the second line of eq. (1.15) trivializes for $k = N/2$. As above, the dualities in the second line in eqs. (1.15) and (1.16) are obtained from the first line by "analytic continuation" to negative $k$ combined with orientation reversal.

Furthermore, these dualities motivate us to conjecture also the phase diagram of these two theories. Therefore, we will also study the theories with gauge group $SO$ ($Sp$) with a fermion in the symmetric (antisymmetric) representation. We will find through our dualities above that the infrared description also contains an intermediate quantum phase, given by the Chern-Simons TQFTs

$$SO\left(\frac{N+2}{2} - k\right)_{\frac{N+2}{2} + k} \qquad \text{for} \quad SO(N)_k + S \quad \text{with} \quad k < \frac{N}{2},$$

$$Sp\left(\frac{N-1}{2} - k\right)_{\frac{N-1}{2} + k} \qquad \text{for} \quad Sp(N)_k + A \quad \text{with} \quad k < \frac{N-1}{2} \quad (1.17)$$

in the deep infrared. An important distinction from the adjoint QCD theories is that now there is no Majorana fermion in the infrared. (These theories do not become supersymmetric for any value of the mass.) Nevertheless, the time-reversal anomaly $\nu$ modulo 16 of the $k = 0$ ultraviolet gauge theory beautifully matches the $T$-reversal anomaly of the intermediate TQFT eq. (1.14)

$$SO\left(\frac{N+2}{2}\right)_{\frac{N+2}{2}} \longleftrightarrow SO\left(\frac{N+2}{2}\right)_{-\frac{N+2}{2}},$$

$$Sp\left(\frac{N-1}{2}\right)_{\frac{N-1}{2}} \longleftrightarrow Sp\left(\frac{N-1}{2}\right)_{-\frac{N-1}{2}}. \quad (1.18)$$

This corroborates our conjectures about the infrared phases of these theories and the dualities.

It is common to couple quantum field theories to various background fields. In our case we can couple them to background gauge fields for the various global symmetries and to the metric. Then, there can be tests of our picture involving the counterterms for these fields. The counterterms for the background gauge fields of the dualities involving $SO(N)$ gauge theories will be presented in [48]. This will test our phase diagram and will allow us to derive similar results for other gauge groups, e.g. $Spin(N)$.

Another test of our proposal arises from the gravitational Chern-Simons counterterm[13]. Starting in the ultraviolet theory we derive the two phases at large positive and large negative mass at weak coupling. So the difference between the coefficients of this term in these two

---

[13]We thank F. Benini for raising this point.

phases is easily computable. Then, we can use the change in this coefficient when we use level/rank duality (using expressions in [3] for $SU$ groups and in [4] for $SO$ and $Sp$ groups) and the change when we go through the phase transitions to check that we find the same answer in the middle phase regardless of whether we arrive to it from positive mass or negative mass. When we perform this check we should make sure that in the theories with a massless Goldstino this coefficient changes as we go through that point. Since this computation is straightforward and tedious and since similar computations were done in [49,50], we will not present it in detail here. We will simply state that this consistency check is satisfied in all the cases we discuss here.

The plan of the paper is as follows. In section 2 we consider in detail $SU(N)$ Chern-Simons theory with an adjoint fermion and discuss some interesting special cases. In section 3 we consider the analogous problem for $SO(N)$ and $Sp(N)$ adjoint QCD. In section 4 we discuss the phase diagram of $SO(N)$ theories with a fermion in the symmetric-traceless representation and $Sp(N)$ theories with a fermion in the antisymmetric-traceless representation.

# 2  $SU(N)$ Adjoint QCD Phase Diagrams

In this section we study the phase diagram of Yang-Mills theory with gauge group $SU(N)$, level $k$ Chern-Simons term and a fermion $\lambda$ in the adjoint representation. The two asymptotic infrared phases that describe the domain of large fermion mass (for any value of $k$) are the TQFTs $SU(N)_{k+N/2}$ and $SU(N)_{k-N/2}$. We discuss the infrared dynamics for $k \geq N/2$ and $k < N/2$ in turn.

## 2.1  Phase Diagram for $k \geq N/2$

The phase diagram can be understood by combining the TQFTs $SU(N)_{k\pm N/2}$ that can be reliably studied at large mass with the expectations from the supersymmetric theory at $m_\lambda = 0$. At $m_\lambda = 0$ the infrared theory is believed to consist simply of $SU(N)_{k-N/2}$ all the way down to $k = N/2$ [14].

A natural guess is that $SU(N)$ adjoint QCD for $k \geq N/2$ has only two phases, described by $SU(N)_{k\pm N/2}$. The supersymmetric theory with $m_\lambda = 0$ is in the phase $SU(N)_{k-N/2}$. This is true also for $k = N/2$, where the supersymmetric theory is in the trivial phase $SU(N)_0$ (and hence has a unique ground state). There is a phase transition between these two phases at some *nonzero* (positive) value of the supersymmetry-breaking mass $m_\lambda$. The phase diagram is depicted in fig. 4. We emphasize again that the phase transition is not coincident with the supersymmetric point.

An interesting special case is $G = SU(2)_k$ with odd $k$. Here we we can gauge the $\mathbb{Z}_2$ one-form symmetry and arrive at $SO(3)_{k/2} = SU(2)_k/\mathbb{Z}_2$ with a single real fermion in the three-dimensional (adjoint) representation. (The supersymmetric point of this theory played in an important role in [14].) This theory was argued in [4,23,43] to be dual to an $SO\left(\frac{k+1}{2}\right)_{-3}$ gauge theory coupled to a scalar in the vector representation at a Wilson-Fisher point. Following [43] (see footnote 3 there) we identify the point where the duality is relevant with the transition point in fig. 4, which differs from the supersymmetric point.

We can recover the $SU(2)_k$ theory by gauging the magnetic $\mathbb{Z}_2$ global symmetry in this duality. This is done in detail in [48] with the conclusion that the $SU(2)_k$ theory coupled to a fermion in the adjoint is dual to $O\left(\frac{k+1}{2}\right)^0_{-3,-3}$ coupled to a scalar in the vector representation. Here the superscript and the two subscripts label various topological terms in the gauge theory. The first subscript is the Chern-Simons level of the continuous group. The superscript denotes a coupling between the continuous group and the discrete $\mathbb{Z}_2$ (it was introduced in [51] and

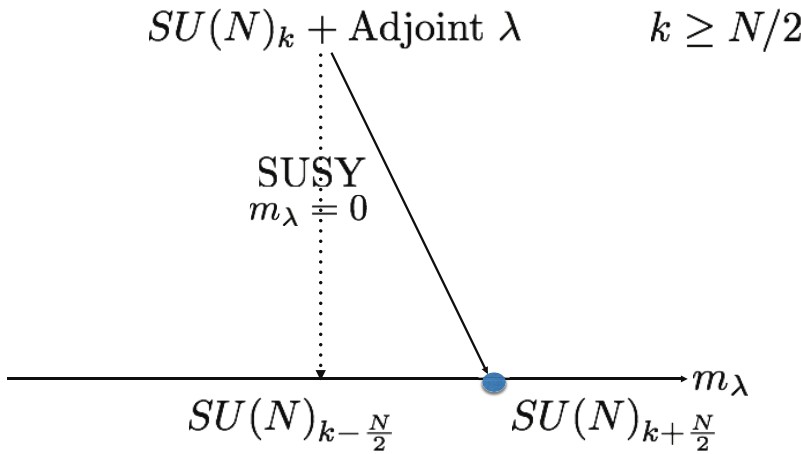

Figure 4: The phase diagram of $SU(N)_k$ with an adjoint fermion for $k \geq N/2$. Note that the physics at the supersymmetric point is smooth. For $k = N/2$ the negative mass phase is trivial. In that case the transition is still to the right of the supersymmetric point.

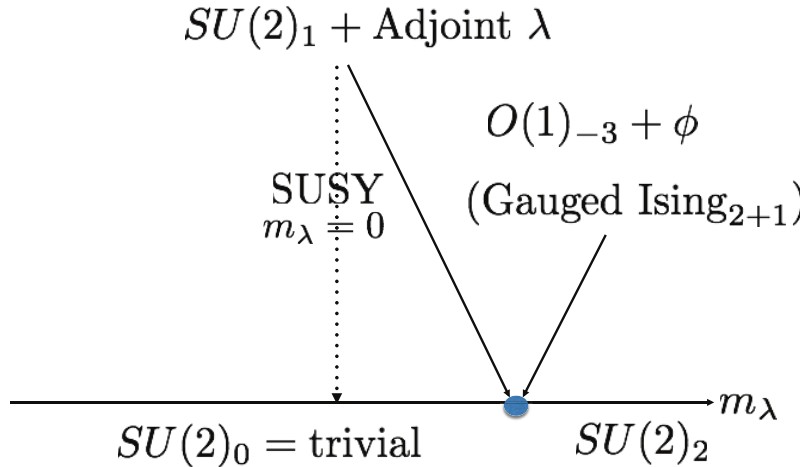

Figure 5: The phase diagram of $SU(2)_1$ with an adjoint fermion. Note that the physics at the supersymmetric point is smooth in this example, which corresponds to $k = N/2$. We also present the dual description. Since $O(1) \sim \mathbb{Z}_2$, the dual description of the transition is in terms of the gauged 2+1d Ising model with a certain topological term in the $\mathbb{Z}_2$ gauge theory.

denoted there by $\pm$). In this case it is 0, which means that this coupling is absent. And the second subscript is a topological term in the $\mathbb{Z}_2$ sector. On a spin manifold such terms have a $\mathbb{Z}_8$ classification[14] [53,54] (see also [55,56]) and the subscript $-3$ means that our term is the third power of the generator of that $\mathbb{Z}_8$. See [48] for more details. The $O\left(\frac{k+1}{2}\right)^0_{-3,-3}$ theory coupled to a scalar in the vector representation has two weakly coupled gapped phases with TQFTs. The TQFT of the phase where the scalar condenses is level/rank dual to an $SU(2)_{k-1}$ TQFT, while the TQFT in the phase where the scalar is massive is level/rank dual to an $SU(2)_{k+1}$ TQFT [48].

Two special cases $k = 1$ and $k = 3$ are particularly simple. For $k = 1$ we have $O(1) \sim \mathbb{Z}_2$,

---

[14]This $\mathbb{Z}_8$ classification of the anomaly is closely related to the standard anomaly in performing a chiral GSO projection in two-dimensional theories. In this form this $\mathbb{Z}_8$ is crucial in the consistency of type II and heterotic string theories [52]. We thank the referee for pointing this out to us.

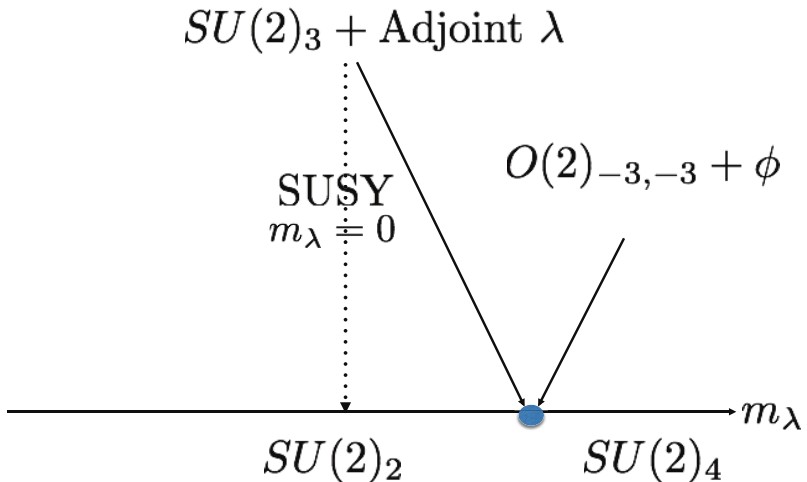

Figure 6: The phase diagram of $SU(2)_3$ with an adjoint fermion. We also present a bosonic dual description – a gauged version of the $O(2)$ Wilson-Fisher point.

so the dual theory is a gauged version of the $2 + 1$ dimensional Ising model [48]. This is depicted in fig. 5. For $k = 3$ the dual theory is $O(2)_{-3,-3}$ coupled to a scalar in the vector representation [48]. This is a gauged version of the $O(2)$ Wilson-Fisher model, where again, the two subscripts denote the topological terms in the gauge field. (In this case the superscript $+$ of the general case is superfluous.) This case is depicted in fig. 6.

## 2.2 Phase Diagram for $k < N/2$

Here we turn to the theory with low $k$. The TQFTs $SU(N)_{k \pm N/2}$ still describe the asymptotic large mass phases of the diagram. Since at $m_\lambda = 0$ the theory is believed to break supersymmetry spontaneously [14], this implies the presence of a Majorana Goldstino particle at the supersymmetric point. Therefore the phase diagram of fig. 4 needs to be somewhat modified. We have already argued in the introduction that a possible, well-motivated, modification of the phase diagram consists of introducing another phase transition so that the system has generically three distinct phases. Two phases are visible semi-classically and are described by Chern-Simons TQFTs and the third phase is a new quantum phase. The new phase cannot consist just of a Goldstino as this would not match various symmetries and anomalies of adjoint QCD. These include the one-form $\mathbb{Z}_N$ symmetry of adjoint QCD and its 't Hooft anomaly and the anomaly in the $T$-reversal symmetry of the $k = 0$ theory. We will discuss this latter anomaly below. We will suggest that the infrared theory in the new phase contains a TQFT (which we identify below) in addition to the Majorana Goldstino.

Recall that at large positive and negative $m_\lambda$ the theory is described in the infrared by $SU(N)_{k \pm \frac{N}{2}}$ Chern-Simons theory, respectively. Using level/rank duality it is useful to rewrite these TQFTs as [3]

$$SU(N)_{k \pm \frac{N}{2}} \longleftrightarrow U\left(\frac{N}{2} \pm k\right)_{\mp N} = \frac{SU\left(\frac{N}{2} \pm k\right)_{\mp N} \times U(1)_{\mp N\left(\frac{N}{2} \pm k\right)}}{\mathbb{Z}_{\frac{N}{2} \pm k}} . \tag{2.1}$$

The equivalence between the theories in crefLR is only as spin-TQFTs [3], but we can use it since adjoint QCD requires a choice of spin structure due to the fermion in the adjoint representation of $SU(N)$.

We suggest that at small $m_\lambda$ the infrared is described by a *different* Chern-Simons TQFT

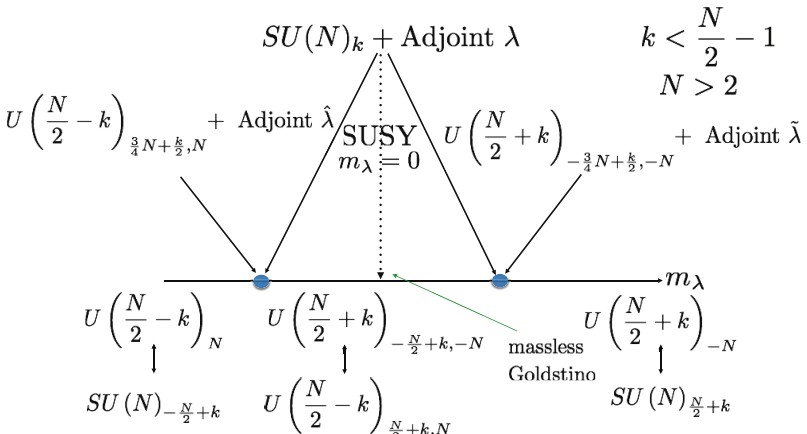

Figure 7: The phase diagram of $SU(N)_k$ with an adjoint fermion and $k < N/2-1$ and $N > 2$. In this theory there are two transitions where the infrared TQFT has to change and in addition one point where the Goldstino becomes massless, in between the two transitions. A fermionic dual for both of the nontrivial transitions is proposed.

(see fig. 7)

$$U\left(\frac{N}{2}+k\right)_{-\frac{N}{2}+k,-N}, \tag{2.2}$$

or, equivalently by level/rank duality [3]

$$U\left(\frac{N}{2}-k\right)_{\frac{N}{2}+k,N}. \tag{2.3}$$

As we said, at the supersymmetric point $m_\lambda = 0$, the infrared theory consists of a Goldstino and a TQFT, which we now have identified with eq. (2.2).

Let us summarize our proposal. Consider $SU(N)$ adjoint QCD with $k < N/2 - 1$. At large positive mass the infrared theory is a Chern-Simons TQFT $SU(N)_{k+N/2} \longleftrightarrow U\left(\frac{N}{2}+k\right)_{-N}$. As we decrease the mass we encounter a transition to the TQFT $U\left(\frac{N}{2}+k\right)_{-\frac{N}{2}+k,-N} \longleftrightarrow U\left(\frac{N}{2}-k\right)_{\frac{N}{2}+k,N}$. As we proceed along the mass axis the Goldstino becomes massless at the supersymmetric point and then massive again. Finally, we encounter the last transition to the $SU(N)_{k-N/2} \longleftrightarrow U\left(\frac{N}{2}-k\right)_{N,N}$ Chern-Simons TQFT.

Let us consider more carefully the first transition, namely, the transition between $SU(N)_{k+\frac{N}{2}} \longleftrightarrow U\left(\frac{N}{2}+k\right)_{-N,-N}$ and $U\left(\frac{N}{2}+k\right)_{-\frac{N}{2}+k,-N}$. This can be nicely reproduced using a dual fermionic theory

$$U\left(\frac{N}{2}+k\right)_{-\frac{3}{4}N+\frac{k}{2},-N} + \text{ adjoint } \tilde{\lambda}, \tag{2.4}$$

where $\tilde{\lambda}$ is the adjoint of $SU\left(\frac{N}{2}+k\right)$ – there is no singlet. The fermion is denoted by $\tilde{\lambda}$ rather than $\lambda$ in order to distinguish it from the original fermion. Since there is no matter charged under the $U(1)$ factor, the phase diagram of this model is that of $SU(\frac{N}{2}+k)_{-\frac{3}{4}N+\frac{k}{2}}$ with an adjoint fermion, $\tilde{\lambda}$. Since in the range $k < N/2$ we have $2|\frac{3}{4}N - \frac{k}{2}| > \frac{N}{2} + k$, we see, according to our previous analysis (for $k \geq N/2$ in the notations of the original theory), that there is one phase transition in this theory and it describes the transition between the two phases we need. Similarly, there is a dual fermionic theory for the transition between $SU(N)_{k-N/2} \longleftrightarrow U\left(\frac{N}{2}-k\right)_{N,N}$

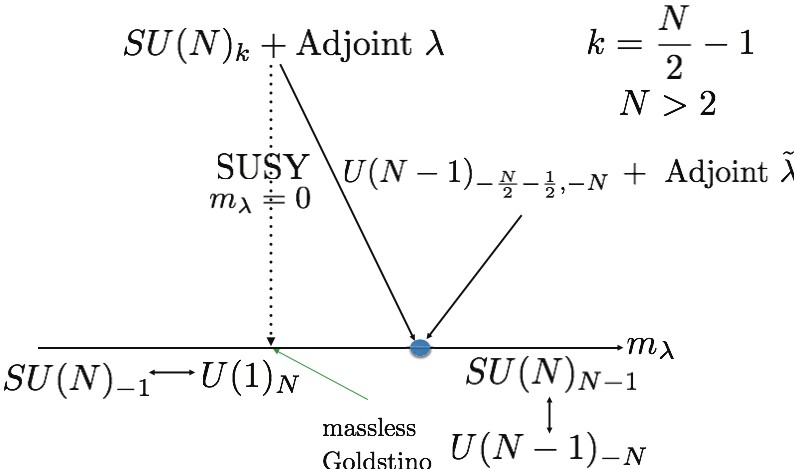

Figure 8: The phase diagram of $SU(N)_k$ with an adjoint fermion and $k = N/2-1$ and $N > 2$. In this theory there is one transition where the infrared TQFT has to change and in addition one point where the Goldstino becomes massless. We propose a fermionic dual for the nontrivial transition.

and $U\left(\frac{N}{2} - k\right)_{\frac{N}{2}+k,N}$, described by adjoint QCD with a fermion $\hat{\lambda}$

$$U\left(\frac{N}{2} - k\right)_{\frac{3}{4}N+\frac{k}{2},N} + \text{adjoint } \hat{\lambda} \, . \tag{2.5}$$

We therefore arrive at two new fermion-fermion dualities

$$SU(N)_k + \text{adjoint } \lambda \longleftrightarrow U\left(\frac{N}{2} + k\right)_{-\frac{3}{4}N+\frac{k}{2},-N} + \text{adjoint } \tilde{\lambda} \, , \tag{2.6}$$

$$SU(N)_k + \text{adjoint } \lambda \longleftrightarrow U\left(\frac{N}{2} - k\right)_{\frac{3}{4}N+\frac{k}{2},N} + \text{adjoint } \hat{\lambda} \, , \tag{2.7}$$

which are valid for $k < \frac{N}{2}$, and describe the two transitions discussed above. As we said above, the second duality follows from the first by "analytic continuation" to negative $k$ combined with orientation reversal.

We would like to clarify a possibly confusing point. $SU(N)_k$ adjoint QCD has an $\mathcal{N} = 1$ supersymmetric point, which we have denoted by $m_\lambda = 0$. Our dual descriptions eq. (2.6) and eq. (2.7) are supposed to describe the phase transitions of $SU(N)_k$ adjoint QCD for $k < N/2$ and away from the supersymmetric point $m_\lambda = 0$. Yet, the dual theories consist of $U$ gauge groups with an adjoint fermion $\tilde{\lambda}, \hat{\lambda}$, and if we added another massive singlet fermion these theories would also have their own $\mathcal{N} = 1$ supersymmetric point. There is no relation between these supersymmetric points and those of the original $SU(N)_k$ adjoint QCD theory. In our $SO$ and $Sp$ dualities in section 3 this point will become even more apparent, as the dual theories do not have an $\mathcal{N} = 1$ supersymmetric point in the first place. There is no contradiction here because the dualities eq. (2.6) and eq. (2.7) describe transitions that are away from the supersymmetric points $m_\lambda = 0$.

An interesting special case occurs for $k = N/2 - 1$. In that case the asymptotic theories at large positive mass and large negative mass are given, respectively, by $U(N-1)_{-N,-N}$ and $U(1)_N$ (see fig. 8). The quantum phase is also given by eq. (2.3), i.e. $U(1)_N$ Chern-Simons theory. Therefore, the Chern-Simons theory at small $m_\lambda$ is identical to one of the asymptotic TQFTs. The transition that is dual to eq. (2.6) remains nontrivial, while the dual theory eq. (2.7) becomes $U(1)_N$ with no matter fields, and hence it is a trivial dual description.

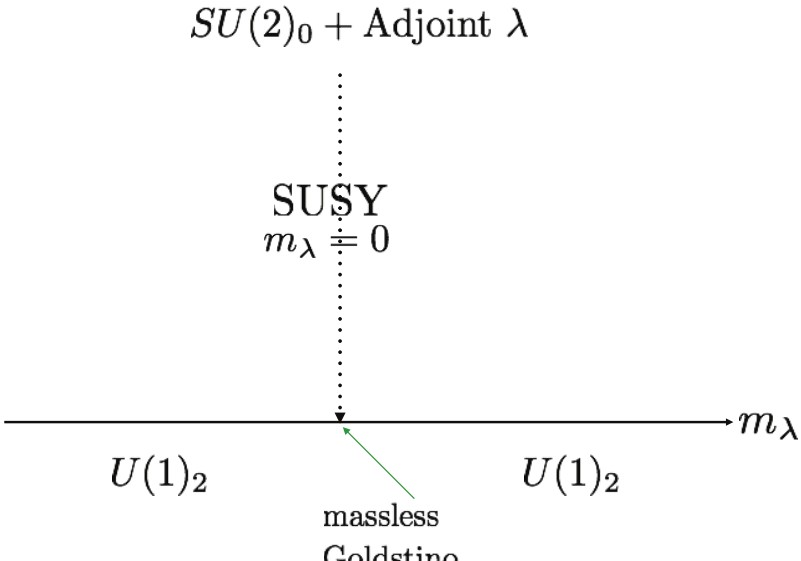

Figure 9: The phase diagram of $SU(2)_0$ with an adjoint fermion. In this diagram there are no necessary transitions other than the Goldstino becoming massless at one point. This is due to the fact that all the following four TQFTs are dual: $SU(2)_{-1} \longleftrightarrow SU(2)_1 \longleftrightarrow U(1)_2 \longleftrightarrow U(1)_{-2}$. The phase diagram therefore is especially simple. It could be summarized by saying that there is a duality between $SU(2)_0$ with an adjoint fermion and a free Majorana fermion accompanied by a pure $U(1)_2$ TQFT.

There is a massless Goldstino somewhere in the part of the phase diagram that is described by a $U(1)_N$ TQFT. In other words, the $\mathcal{N} = 1$ theory with $m_\lambda = 0$ flows to a $U(1)_N$ TQFT accompanied by a massless Goldstino.

For $N = 2$, the theory with $k = N/2 - 1 = 0$ is time-reversal invariant at $m_\lambda = 0$ and the picture is further simplified since the asymptotic phase at large positive $m_\lambda$ is $U(1)_{-2}$ (see fig. 9). There is a massless Goldstino at $m_\lambda = 0$ and the TQFT at $m_\lambda = 0$ can be chosen to be $U(1)_2$ or $U(1)_{-2}$, which are identical by level/rank duality. Therefore, we see that in $SU(2)_0$ with an adjoint fermion there is a single second-order transition at $m_\lambda = 0$. At the second order transition point, the Goldstino becomes massless. Therefore, in this case one can summarize the situation by the statement that there is a duality

$$SU(2)_0 + \text{adjoint } \lambda \longleftrightarrow \text{neutral } \psi + U(1)_2 \,, \tag{2.8}$$

with $\psi$ a neutral Majorana fermion. This duality is reminiscent of the supersymmetric duality [57].

The $SU(2)_0$ adjoint QCD that we have just discussed is also a special member of the $T$-invariant family of theories $SU(N)_0$ adjoint QCD, which exhibit three phases for $N > 2$ (see fig. 10). We will discuss these $T$-invariant theories below.

## 2.3 Symmetries and 't Hooft Anomalies Matching

As with all dualities and proposed infrared behavior of strongly coupled theories, one has to check that the symmetries of the dual theories and their anomalies match. As we said in the introduction, most of this is guaranteed to work in our setup. We used a sequence of steps: integration out of the fermion at large positive mass, level/rank duality, a transition described by the weakly coupled theory eq. (2.6), level/rank duality, a transition described by the weakly coupled theory eq. (2.7), and level/rank duality. This matched with the integration of the

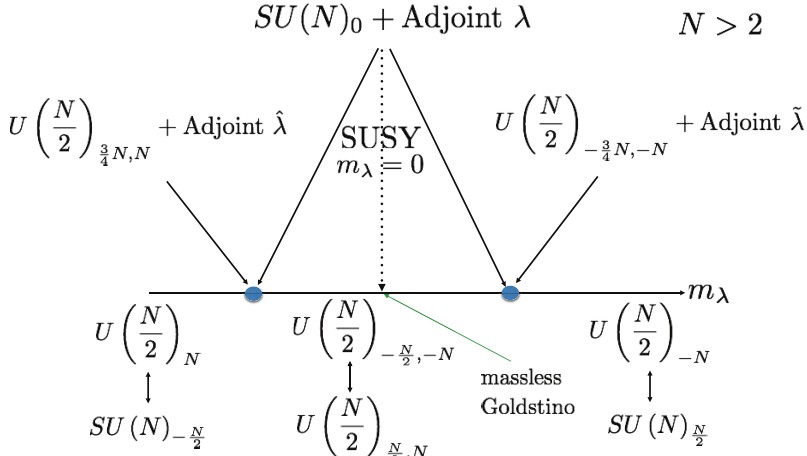

Figure 10: This is the time-reversal invariant theory with an adjoint fermion. The $\mathcal{N} = 1$ supersymmetric point coincides with the time-reversal invariant point. The theory flows to a massless Goldstino and a $U(N/2)_{N/2,N}$ TQFT.

fermion at large negative mass. Every step here either involves a computation in a weakly coupled theory, or level/rank duality, which is rigorously proven. This does not prove that our proposed phase diagram is correct. But it does show that all the phases we described have the same global symmetries and they have the same 't Hooft anomalies.

There is an exception to this reasoning. For $k = 0$ (which is possible only for $N$ even) our system is $T$-reversal invariant for $m_\lambda = 0$, but it is not invariant for nonzero $m_\lambda$. Since our argument for the global symmetry and its anomaly matching between the ultraviolet theory and the infrared theory involved first making $m_\lambda$ nonzero, there is no guarantee that our proposed infrared theory for $k = m_\lambda = 0$ is $T$-reversal invariant. And even if it is invariant, there is no guarantee that the 't Hooft anomaly in this symmetry in the ultraviolet matches that of the infrared.

Regardless of our specific proposal, the infrared behavior of the $k = m_\lambda = 0$ theory either has to be time-reversal preserving or the time-reversal symmetry must be spontaneously broken. The latter is excluded by the Vafa-Witten theorem [26]. In our proposal (fig. 10) the infrared theory consists of a massless Majorana fermion $\psi$ and the $U\left(\frac{N}{2}\right)_{\frac{N}{2},N}$ TQFT. The massless Majorana fermion is manifestly time-reversal invariant, while the fact that $U\left(\frac{N}{2}\right)_{\frac{N}{2},N}$ is a time-reversal invariant TQFT follows from level/rank duality even though this is not a manifest symmetry of the Lagrangian [4].

The matching of the 't Hooft anomaly in the time-reversal symmetry at $k = m_\lambda = 0$ is not obvious. This time-reversal anomaly is related to the eta invariant in 3+1 dimensions and it is an integer $\nu$ modulo 16. This anomaly in adjoint QCD is easily calculated using the $N^2 - 1$ fermions. (At very short distances the gauge fields decouple and the theory is essentially a theory of free fermions and free gauge fields. Therefore, only the fermions contribute to the anomaly.) It is

$$\nu_{UV} = (N^2 - 1)\mathrm{mod}\,16 = \left(1 - 2(-1)^{N/2}\right)\mathrm{mod}\,16 \tag{2.9}$$

(recall that $N$ has to be even for $k = 0$).[15] In general the value of $\nu_{IR}$ in our infrared theory is not easy to compute, see for instance [39, 44, 45, 59, 60]. One contribution to it, due to

---

[15]Given a Majorana fermion, its contribution to $\nu$ can be $\pm 1$ depending on the action of $T$ on that fermion. Above we have assumed that $T$ acts in the same way on all the $N^2 - 1$ Majorana fermions. However, one still has to check that this prescription is gauge-independent since we may compose an $SU(N)$ gauge transformation with the action of $T$ and change the way $T$ acts on some of the Majorana fermions. Indeed, consider an $SU(N)$

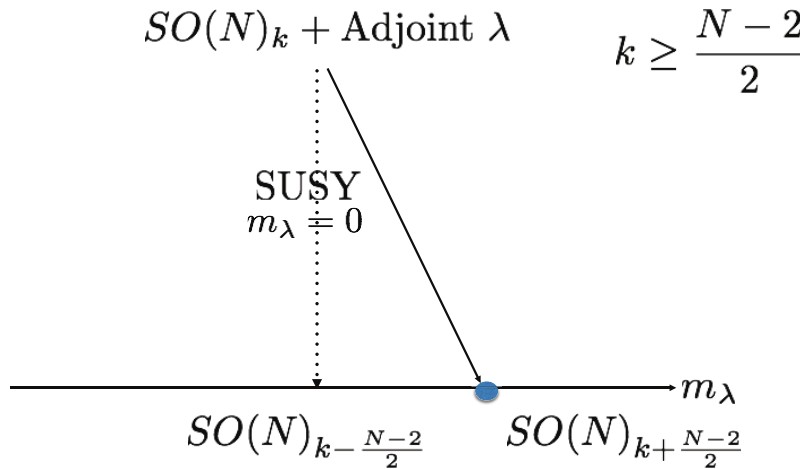

Figure 11: The phase diagram for $SO(N)_k$ gauge theory with an adjoint fermion for $k \geq (N-2)/2$. The physics at the supersymmetric point is smooth. For $k = (N-2)/2$ the negative mass phase is trivial. In that case the transition is still to the right of the supersymmetric point.

the Goldstino, is 1.[16] The contribution of the TQFT $U\left(\frac{N}{2}\right)_{\frac{N}{2},N}$ was worked out in [44–46] (see also references therein) and was found to be $\pm 2$. We suggest that in the present context time-reversal in the infrared would act such that it is actually $-2(-1)^{N/2}$. Therefore, we find $\nu_{IR} = 1 - 2(-1)^{N/2}$, as in the ultraviolet!

## 3 Phase Diagrams and Dualities for $SO(N)$ and $Sp(N)$

In this section we study the phase diagram of adjoint QCD with $SO$ and $Sp$ gauge groups. We often denote the gauge group by $G$ when it makes no difference which of the two cases we are discussing. We discuss the long distance behavior of these theories as a function of the mass $m_\lambda$ of the fermion and of $k$. Our discussion will be along the lines of the general description in the introduction and will be quite similar to the analysis of $SU$ theories above.

For any value of $k$, the phase diagram has two semiclassically accessible phases where the fermion is very massive. These are described by Chern-Simons theories $G_{k+h/2}$ and $G_{k-h/2}$ respectively, where $h = N - 2$ for $SO(N)$ and $h = N + 1$ for $Sp(N)$. These TQFTs describe the asymptotics of the phase diagram. We now proceed to complete the phase diagram for $k \geq h/2$ and $k < h/2$.

---

gauge transformation $g = \mathrm{diag}(-1, ..., -1, 1, .., 1)$ with $2k$ entries of $-1$. Then there are two groups of Majorana fermions: one of size $4k^2 + (N-2k)^2 - 1$ and one of size $4k(N-2k)$. The orientation of the action of $T$ on the second group is the opposite of the orientation of the action on the first group. The anomaly is therefore

$$\nu = \left(4k^2 + (N-2k)^2 - 1 - 4k(N-2k)\right)\mathrm{mod}16 = (N^2 - 1)\mathrm{mod}16 \,, \tag{2.10}$$

where we used the fact that $N$ is even. We see that our prescription for computing $\nu$ in the ultraviolet is unambiguous. For a related discussion see [41, 58].

[16]The sign of the contribution of the Goldstino can be understood from the action of time-reversal on the supercurrent $Tr(F\lambda)$, which interpolates to the Goldstino in the deep infrared as $G_\alpha \sim Tr(F\lambda)$. This shows that the correct sign in the infrared is $+1$. The same argument holds for the other gauge groups we discuss later.

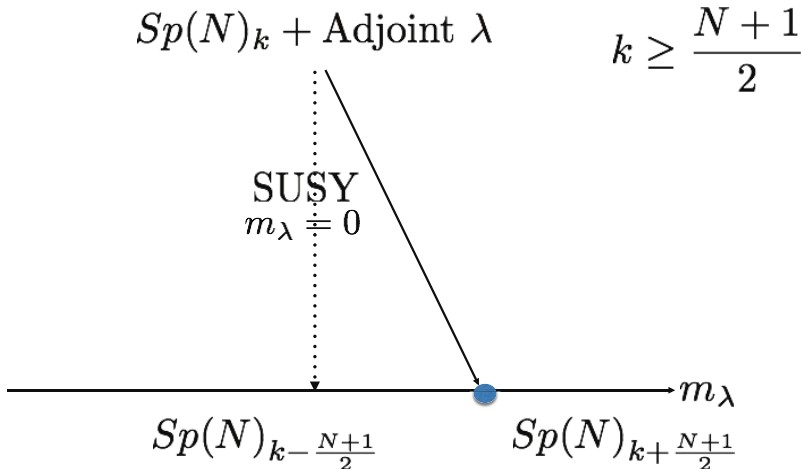

Figure 12: The phase diagram for $Sp(N)_k$ gauge theory with an adjoint fermion for $k \geq (N+1)/2$. The physics at the supersymmetric point is smooth. For $k = (N+1)/2$ the negative mass phase is trivial. In that case the transition is still to the right of the supersymmetric point.

## 3.1 Phase Diagram for $k \geq h/2$

At the supersymmetric point $m_\lambda = 0$ the infrared description can be reliably obtained at large $k$ by integrating out $\lambda$ and yields the Chern-Simons theory $G_{k_{IR}}$ based on the gauge group $G$ with level $k_{IR} = k - h/2$. This infrared description has been argued in [14] to be applicable beyond the large $k$ regime all the way down to $k = h/2$. This leads to a rather simple phase diagram. There are two asymptotic phases described by $G_{k-h/2}$ and $G_{k+h/2}$ separated by a phase transition. The supersymmetric point is inside the phase described by $G_{k-h/2}$. The physics at the supersymmetric point is completely regular and the phase transition occurs to the right of the supersymmetric point in the phase diagram. For $k = h/2$ one phase is trivial, labeled by $G_0$ which meets the nontrivial phase $G_h$ at a phase transition. The phase diagrams for $SO$ and $Sp$ for $k \geq h/2$ are summarized in fig. 11 and fig. 12.

## 3.2 Phase Diagram for $k < h/2$

The modification of this picture for $k = h/2 - 1$ is straightforward and is similar to $SU(N)_{\frac{N}{2}-1}$ in fig. 8. $SO(N)_{\frac{N-2}{2}-1}$ has a single phase transition separating $SO(N)_{-1}$ (which is trivial) and $SO(N)_{N-3}$.[17] And $Sp(N)_{\frac{N+1}{2}-1}$ has a single phase transition separating $Sp(N)_{-1}$ and $Sp(N)_N$. Supersymmetry is spontaneously broken in this case [14] with a massless Goldstino point in the $G_{-1}$ phase. This is depicted in fig. 13 and fig. 14. We will return to this special case below.

As in our discussions above, this simple phase diagram needs to be modified for $k < \frac{h}{2} - 1$ and we propose that for $k < h/2 - 1$ there are two phase transitions connecting the two asymptotic phases that are visible semiclassically to an intermediate, inherently quantum mechanical, phase. The supersymmetric theory is in the intermediate phase.

Let us consider an $SO(N)$ gauge theory with Chern-Simons level $k < h/2$ and an adjoint fermion $\lambda$. The infrared Chern-Simons theory at large positive mass is $SO(N)_{k+\frac{N-2}{2}} \longleftrightarrow SO\left(\frac{N-2}{2} + k\right)_{-N}$. As the mass is decreased we cross a phase transition and

---

[17]As we said above, our discussion will not apply to the special case with gauge group $SO(4)$, where the low energy theory includes two Goldstinos, one from each of the $SU(2)$ sectors of the theory.

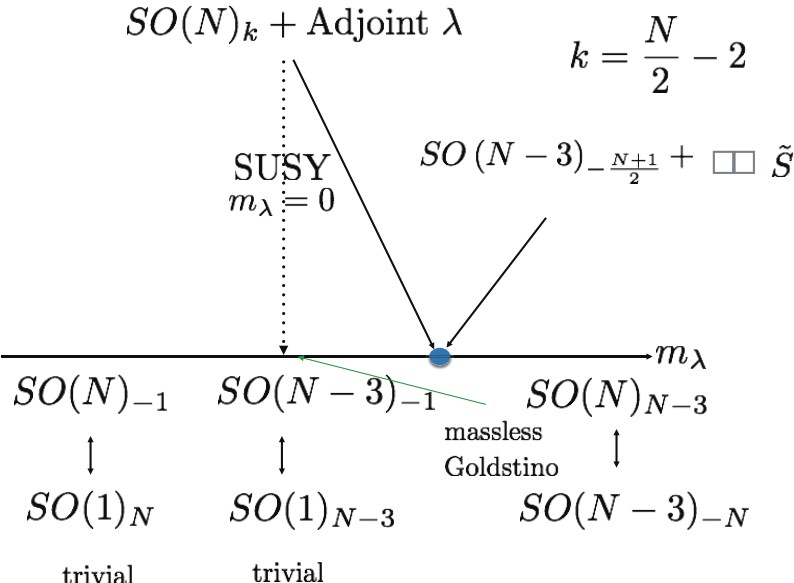

Figure 13: The phase diagram of $SO(N)_k$ gauge theory with an adjoint fermion and Chern-Simons level $k = \frac{N}{2} - 2$. There is one transition that connects a nontrivial TQFT and a trivial one. In addition at one point the Goldstino becomes massless. We propose a fermionic dual for the nontrivial transition.

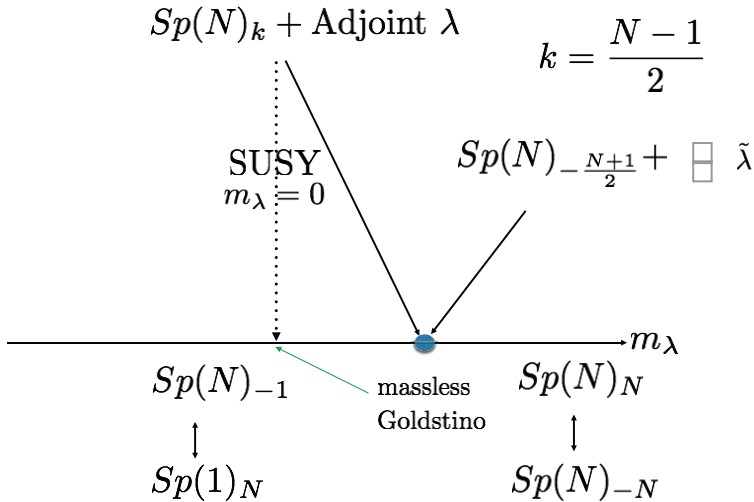

Figure 14: The phase diagram of $Sp(N)_k$ gauge theory with an adjoint fermion and Chern-Simons level $k = \frac{N-1}{2}$. There is one transition that connects two nontrivial TQFTs. In addition at one point the Goldstino becomes massless. We propose a fermionic dual for the nontrivial transition.

encounter an intermediate phase described by a distinct Chern-Simons theory

$$SO\left(\frac{N-2}{2} + k\right)_{-\frac{N-2}{2}+k} \longleftrightarrow SO\left(\frac{N-2}{2} - k\right)_{\frac{N-2}{2}+k}. \qquad (3.1)$$

The supersymmetric point, where there is also a massless Goldstino occurs in this phase.

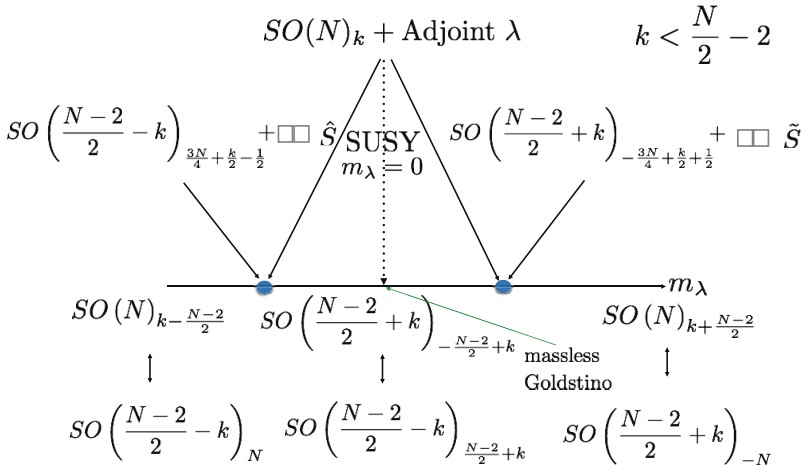

Figure 15: The phase diagram for $SO(N)_k$ gauge theory with an adjoint fermion for $k < N/2 - 2$. There are two phase transitions between different infrared TQFTs. There is also a massless Goldstino at the supersymmetric point. We propose a dual fermionic description of the two transitions.

|  | $SO(N)$ | $Sp(N)$ |
|---|---|---|
| antisymmetric | $N-2$ | $N-1$ |
| symmetric | $N+2$ | $N+1$ |

Table 4: $T(R)$ for the symmetric and antisymmetric representations.

As we decrease the mass further we encounter another phase transition to the asymptotic theory $SO(N)_{k-\frac{N-2}{2}} \longleftrightarrow SO\left(\frac{N-2}{2} - k\right)_N$. The phase diagram is summarized in fig. 15.

The transition between the left and intermediate phase can be reproduced by an $SO\left(\frac{N-2}{2} - k\right)$ gauge theory with a Chern-Simons term at level $\frac{3N}{4} + \frac{k}{2} - \frac{1}{2}$ and a Majorana fermion $\hat{S}$ in the symmetric-traceless representation of the gauge group. Giving a mass to $\hat{S}$ allows us to integrate it out and obtain the two infrared TQFTs neighbouring the transition.[18]

Likewise, the transition between the right and intermediate phase can be reproduced by an $SO\left(\frac{N-2}{2} + k\right)$ gauge theory with a Chern-Simons term at level $-\frac{3N}{4} + \frac{k}{2} + \frac{1}{2}$ and a fermion $\tilde{S}$ in the symmetric-traceless representation of the gauge group.

This suggests the fermion-fermion dualities:

$$SO(N)_k + \text{adjoint } \lambda \longleftrightarrow SO\left(\frac{N-2}{2} + k\right)_{-\frac{3N}{4} + \frac{k}{2} + \frac{1}{2}} + \text{symmetric } \tilde{S} \,,$$

$$SO(N)_k + \text{adjoint } \lambda \longleftrightarrow SO\left(\frac{N-2}{2} - k\right)_{\frac{3N}{4} + \frac{k}{2} - \frac{1}{2}} + \text{symmetric } \hat{S} \,. \tag{3.3}$$

Here $\tilde{S}$ and $\hat{S}$ are symmetric-traceless representations. These dualities hold for $k < \frac{N-2}{2}$. As above, these two dualities are related by "analytic continuation" to negative $k$ combined with orientation reversal.

Further consistency checks involving the counter-terms for the background gauge fields are

---

[18]Integrating out a massive Majorana fermion $\psi$ in a representation $R$ of a gauge group $G$ shifts the level of the Chern-Simons term by

$$k \to k + \text{sgn}(M_\psi) \frac{T(R)}{2} \,, \tag{3.2}$$

where $T(R)$ is the index of $G$ in the representation $R$ (see table 4).

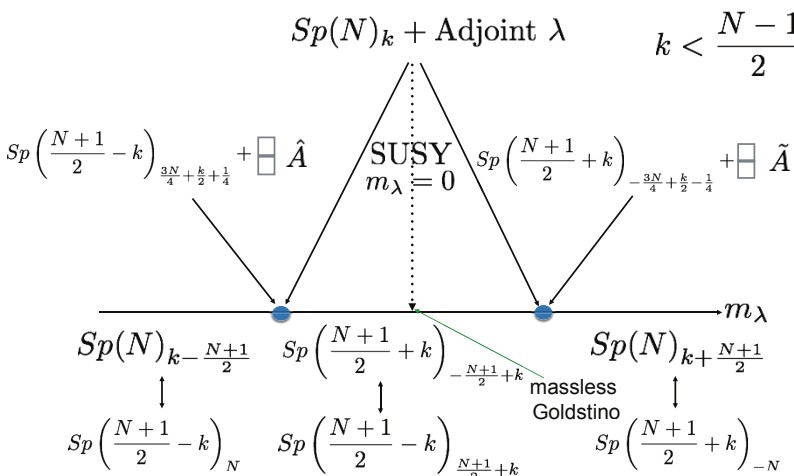

Figure 16: The phase diagram for $Sp(N)_k$ gauge theory with an adjoint fermion for $k < (N-1)/2$. There are two phase transitions between different infrared TQFTs. There is also a massless Goldstino at the supersymmetric point. We propose a dual fermionic description of the two transitions.

discussed in [48, 58]. This allows a determination of the low energy TQFT in other theories, e.g. $Spin(N)_k$ with an adjoint fermion.

A similar picture emerges for $Sp(N)_k$ with an adjoint fermion $\lambda$. There is a transition between the large positive mass $Sp(N)_{k+\frac{N+1}{2}} \longleftrightarrow Sp\left(\frac{N+1}{2}+k\right)_{-N}$ phase and an intermediate phase described by

$$Sp\left(\frac{N+1}{2}+k\right)_{-\frac{N+1}{2}+k} \longleftrightarrow Sp\left(\frac{N+1}{2}-k\right)_{\frac{N+1}{2}+k}. \tag{3.4}$$

The supersymmetric theory flows to this intermediate TQFT with a massless Goldstino, which becomes massive as we depart from the supersymmetric point in the phase diagram. Decreasing the mass further leads to another phase transition to the asymptotic theory $Sp(N)_{k-\frac{N+1}{2}} \longleftrightarrow Sp\left(\frac{N+1}{2}-k\right)_N$. The phase diagram is summarized in fig. 16.

The transition between the left phase and the intermediate phase can be reproduced by an $Sp\left(\frac{N+1}{2}-k\right)$ gauge theory with a Chern-Simons term at level $\frac{3N}{4}+\frac{k}{2}+\frac{1}{4}$ and a fermion $\hat{A}$ in the antisymmetric-traceless representation of the gauge group. Likewise, the transition between the right phase and the intermediate phase can be reproduced by an $Sp\left(\frac{N+1}{2}+k\right)$ gauge theory with a Chern-Simons term at level $-\frac{3N}{4}+\frac{k}{2}-\frac{1}{4}$ and a fermion $\tilde{A}$ in the antisymmetric representation of the gauge group. Using the formula eq. (3.2) for the shift of the Chern-Simons level induced by integrating out a massive fermion we reproduce the asymptotic infrared Chern-Simons theories (see table 4).

This suggests the fermion-fermion dualities:

$$Sp(N)_k + \text{adjoint } \lambda \longleftrightarrow Sp\left(\frac{N+1}{2}+k\right)_{-\frac{3N}{4}+\frac{k}{2}-\frac{1}{4}} + \text{antisymmetric } \tilde{A},$$

$$Sp(N)_k + \text{adjoint } \lambda \longleftrightarrow Sp\left(\frac{N+1}{2}-k\right)_{\frac{3N}{4}+\frac{k}{2}+\frac{1}{4}} + \text{antisymmetric } \hat{A}. \tag{3.5}$$

Here $\tilde{A}$ and $\hat{A}$ are antisymmetric-traceless representations. These dualities hold for $k < \frac{N+1}{2}$. Again, these two dualities are related.

We have already mentioned the case $k = h/2 - 1$ at the beginning of this subsection. This special case can be understood also as we move up from lower values of $k$. Here the infrared

TQFTs that describe the left and intermediate phase become identical. This implies that for $k = h/2 - 1$ there is a single transition between the TQFT at large positive mass governed by the Chern-Simons theory $G_{h-1}$ and another TQFT that governs the rest of the phase diagram. For $SO(N)_{h/2-1}$ adjoint QCD both the left and the intermediate TQFTs become trivial spin-TQFTs (see fig. 13), since $SO(n)_1$ is a trivial spin-TQFT. Therefore, for $SO(N)_{h/2-1}$ adjoint QCD the infrared theory at the supersymmetric point is just a massless Goldstino, without an extra topological sector. The transition admits a dual description given in the first line of eq. (3.3).

For the $Sp(N)$ gauge theory both the left and the intermediate TQFTs become $Sp(1)_N \longleftrightarrow Sp(N)_{-1}$ (see fig. 14). Therefore, for $Sp(N)_{h/2-1}$ adjoint QCD the infrared theory at the supersymmetric point is a massless Goldstino with the Chern-Simons theory $Sp(N)_{-1}$. The transition admits a dual description given in the first line of eq. (3.5).

### 3.3 $T$-reversal symmetry

The adjoint QCD theories with gauge group $SO(N)_k$ and $Sp(N)_k$ with an adjoint fermion are time-reversal invariant when $k = 0$ and $m_\lambda = 0$. This is the supersymmetric point for $k = 0$. This is possible for $G = SO(N)$ only for even $N$ and for $G = Sp(N)$ only for odd $N$ (see table 2). In our scenario the infrared theory consists of a Goldstino, which is time-reversal invariant, and a nontrivial time-reversal invariant TQFT: $SO\left(\frac{N-2}{2}\right)_{\frac{N-2}{2}}$ and $Sp\left(\frac{N+1}{2}\right)_{\frac{N+1}{2}}$ respectively. These TQFTs are time-reversal invariant by virtue of level/rank duality among spin-TQFTs [4]: $SO\left(\frac{N-2}{2}\right)_{\frac{N-2}{2}} \longleftrightarrow SO\left(\frac{N-2}{2}\right)_{-\frac{N-2}{2}}$ and $Sp\left(\frac{N+1}{2}\right)_{\frac{N+1}{2}} \longleftrightarrow Sp\left(\frac{N+1}{2}\right)_{-\frac{N+1}{2}}$. The fact that we find a time-reversal invariant theory in the infrared is a nontrivial consistency check of our proposal.

We would like to analyze the $T$-reversal 't Hooft anomaly in this theory. We start with the $Sp(N)_0$ adjoint QCD following the discussion of $SU(N)$ in section 2. In the ultraviolet the adjoint fermions contribute $\nu_{UV} = N(2N + 1) \bmod 16 = (N + 2) \bmod 16$, where we used the fact that $N$ is odd. (As in footnote 14, it is possible to verify that this prescription for computing $\nu$ is gauge invariant.) In the infrared the Goldstino contributes $+1$ (it is $+1$ rather than $-1$ for the same reason as in section 2). The contribution of the TQFT $Sp\left(\frac{N+1}{2}\right)_{\frac{N+1}{2}}$ to $\nu_{IR}$ follows from $\nu(Sp(n)_n) = \pm 2n \bmod 16$ [47], generalizing $\nu(SU(2)_1) = \pm 2 \bmod 16$ [34] and $\nu(Sp(2)_2) = \pm 4 \bmod 16$ [61]. With an appropriate sign choice for the infrared TQFT we have $\nu_{IR} = (N + 2) \bmod 16$, as in the ultraviolet. This matching is a highly nontrivial test of our phase diagram.

Next, we move to the $T$-reversal 't Hooft anomaly in the $SO(N)_0$ adjoint QCD theory. It exists only for even $N$. Here we use $\nu(SO(n)_n) = \pm n \bmod 16$ [47], generalizing $\nu(SO(2)_2) = \pm 2 \bmod 16$ and $\nu(SO(3)_3) = \pm 3 \bmod 16$ [59].

For $N = 2 \bmod 4$ we have $\nu_{UV} = \frac{N(N-1)}{2} \bmod 16 = \left(-\frac{N}{2} + 2\right) \bmod 16$, which matches the infrared contribution with the sign choice $-\frac{N-2}{2}$ from the TQFT and $+1$ (as above) from the Goldstino.

The situation for $N = 0 \bmod 4$ is more subtle. Here we claim that the naive time-reversal symmetry $T$ of the ultraviolet theory is not mapped to the naive time-reversal symmetry of the infrared theory. Instead, the relevant symmetry in the ultraviolet, whose anomaly we match is $CT$ [58]; i.e. the product of the naive time-reversal symmetry and charge conjugation $C$. The latter acts on the fermions by reversing the sign of the fermions $\lambda^{[1,i]} \to -\lambda^{[1,i]}$ and not changing the sign of the other fermions. This leads to $\nu_{UV} = \left(\frac{(N-1)(N-2)}{2} - (N-1)\right) \bmod 16 = \left(-\frac{N}{2} + 2\right) \bmod 16$. This matches $\nu_{IR}$, which is the sum of $+1$ from the Goldstino and $-\left(\frac{N-2}{2}\right)$ from the TQFT. Again, this is a nontrivial test of our proposal. (One can verify, as in footnote 14, that for $N = 2 \bmod 4$ the anomaly of $T$ is gauge invariant and for $N = 0 \bmod 4$ the anomaly of $CT$ is gauge invariant. For $N = 2 \bmod 4$ the anomaly of $CT$ is not gauge invariant and for $N = 0 \bmod 4$ the anomaly of $T$ is not gauge invariant. For a discussion of the implications of that see [41] and [58].)

### 3.4 Consistency Checks for Low-Rank Theories

The isomorphism of $SO$ and $Sp$ Lie groups for low rank with other Lie groups can lead to further consistency checks of our proposal for the infrared dynamics of these theories. The asymptotic phases of theories related by a Lie group isomorphism are guaranteed to match, since these can be obtained semiclassically. Thus, a further nontrivial check of the phase diagram can be made only when there is an intermediate phase, i.e. for $k < h/2 - 1$.

Our phase diagrams for $Sp(N)$ gauge theories are applicable for all $N$. Here we could look for tests based on $Sp(1) \simeq SU(2)$ and $Sp(2)/\mathbb{Z}_2 \simeq SO(5)$. However, in these cases there is no intermediate phase and therefore there is no non-trivial test.

Our phase diagrams for $SO(N)$ gauge theories are applicable for all $N \neq 1, 2$ and 4. For $N = 1$ there is no gauge group and for $N = 2$ the adjoint fermion is free. For $N = 4$ the adjoint representation is reducible and up to a $\mathbb{Z}_2$ quotient the theory factorizes to two copies of $SU(2)$ with an adjoint.

The group isomorphism $SO(6) \simeq SU(4)/\mathbb{Z}_2$ leads to a nontrivial consistency check only for $k = 0$, where it has an intermediate phase. We first note that even though the $\mathbb{Z}_4$ one-form global symmetry of $SU(4)_k$ has an 't Hooft anomaly for $k \neq 0 \bmod 4$, its $\mathbb{Z}_2$ subgroup is anomaly free since the spin of that line is half-integer. This implies that the quotient Chern-Simons theory $SU(4)_k/\mathbb{Z}_2$ can be defined for any $k$ (on a spin manifold). The intermediate phase of the $SU(4)_0$ gauge theory is described by the TQFT (see fig. 10)

$$U(2)_{2,4} = \frac{SU(2)_2 \times U(1)_8}{\mathbb{Z}_2} \,, \tag{3.6}$$

while the TQFT describing the infrared dynamics in the intermediate phase of $SO(6)_0$ is $SO(2)_2 = U(1)_2$ (see fig. 15). The group isomorphism $SO(6) \simeq SU(4)/\mathbb{Z}_2$ requires gauging a $\mathbb{Z}_2$ subgroup of the $\mathbb{Z}_4$ one-form global symmetry of $U(2)_{2,4}$. This leads to

$$\left( \frac{SU(2)_2 \times U(1)_8}{\mathbb{Z}_2} \right)/\mathbb{Z}_2 \simeq \frac{SU(2)_2}{\mathbb{Z}_2} \times \frac{U(1)_8}{\mathbb{Z}_2} \longleftrightarrow U(1)_2 \,, \tag{3.7}$$

where we used the fact that the first factor $SO(3)_1$ is a trivial spin-Chern-Simons theory and the second factor is $U(1)_2$. This matches the intermediate region of the $SO(6)_0$ gauge theory.

## 4 Phase Diagrams and Dualities for $SO(N)$ with Fermions in the Symmetric Tensor and $Sp(N)$ with Fermions in the Antisymmetric Tensor

In this section we analyze the phase diagram of $SO(N)_k$ with fermions in the symmetric-traceless tensor representation and of $Sp(N)_k$ with fermions in the antisymmetric-traceless tensor representation.[19] We denote a symmetric-traceless fermion of $SO$ by $S$ and by $A$ an antisymmetric-traceless fermion of $Sp$. These theories have already appeared in our proposed dual description of the transitions of $SO/Sp$ adjoint QCD for $k < h/2$ (see eqs. (1.11) and (1.12)).[20] The analysis of this section provides further consistency checks on the previously discussed dualities and gives rise to new ones.

The phase diagrams for these theories resemble those of adjoint QCD. For large positive and negative mass and any value of $k$ there are the two semiclassical phases described by Chern-Simons theory $G_{k+T(R)/2}$ and $G_{k-T(R)/2}$ respectively.

---

[19]We recall that the adjoint of $SO/Sp$ is the rank-two antisymmetric/symmetric representation.

[20]Note that $k$ in this expression and in the previous sections is the level of the theory with adjoint fermions. In most of this section we label by $k$ the level of the fermionic dual of that theory. We hope that this change in notation will not cause confusion.

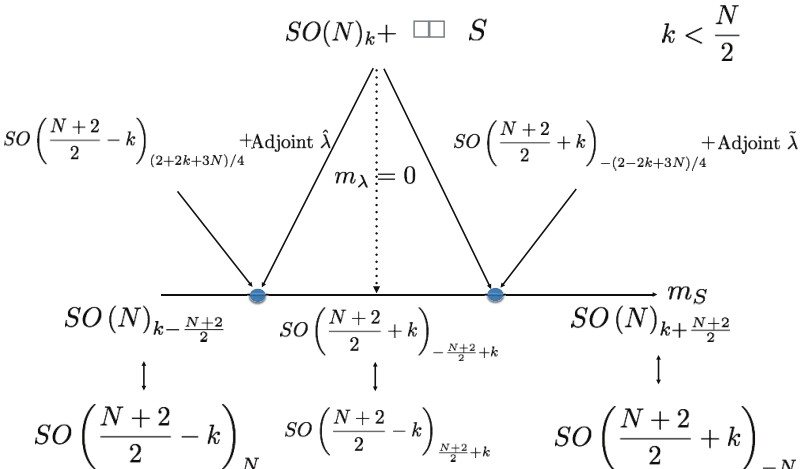

Figure 17: The phase diagram of $SO(N)_k$ gauge theory with a fermion in a symmetric-traceless tensor for $k < N/2$. There are two phase transitions between different infrared TQFTs. We propose a dual fermionic description of the two transitions.

We propose that for $k \geq T(R)/2$ there are two asymptotic phases $G_{k+T(R)/2}$ and $G_{k-T(R)/2}$ connected via a transition. Intuitively, it is at the point where the ultraviolet fermion becomes massless. While this statement can be reliably established for $k \gg 1$, we suggest that this conclusion can be continued all the way down to $k = T(R)/2$, where the theory is strongly coupled (for $k = T(R)/2$ the negative mass phase is trivial). This scenario passes a nontrivial consistency check as these theories were proposed to provide a dual description of the transitions of adjoint QCD for $k < h/2$. Our dualities require that the dual theories in eq. (1.11) and eq. (1.12) have only two phases when $k < h/2$. (Again, $k$ here is that of the gauge theory with adjoint fermions.) And indeed it is simple to verify that this is the case if $SO(N)_k$ with a fermion in a symmetric-traceless tensor $S$ and $Sp(N)_k$ with a fermion in an antisymmetric-traceless tensor $A$ have two phases for $k \geq T(R)/2$.

We now consider the phase diagram for $k < T(R)/2$. In $SO(N)_k$ with a fermion $S$ with $k < T(R)/2 - 1 = N/2$ there is an intermediate phase, which is not visible semiclassically (see fig. 17).[21] This new phase is described by the nontrivial intermediate TQFT

$$SO\left(\frac{N+2}{2}+k\right)_{-\frac{N+2}{2}+k} \longleftrightarrow SO\left(\frac{N+2}{2}-k\right)_{\frac{N+2}{2}+k}. \tag{4.1}$$

The transitions from the semiclassical phases to this quantum phase admit dual fermionic descriptions

$$SO(N)_k + \text{symmetric } S \longleftrightarrow SO\left(\frac{N+2}{2}+k\right)_{-\frac{3N}{4}+\frac{k}{2}-\frac{1}{2}} + \text{adjoint } \tilde{\lambda},$$

$$SO(N)_k + \text{symmetric } S \longleftrightarrow SO\left(\frac{N+2}{2}-k\right)_{\frac{3N}{4}+\frac{k}{2}+\frac{1}{2}} + \text{adjoint } \hat{\lambda}. \tag{4.2}$$

Again, these two dualities are related.

In the theory with $k = T(R)/2 - 1 = N/2$ the negative mass phase and the intermediate phase are trivial and there is a single nontrivial transition (see fig. 18) with a dual description

---

[21] For $N = 2$ the theory is $U(1)_0$ with a fermion of charge two. Here the $\mathbb{Z}_2$ magnetic symmetry is enhanced to a global $U(1)$ symmetry and the intermediate phase shrinks to a point – the infrared behavior of the massless theory includes a free Dirac fermion and a $U(1)_2$ TQFT [58]. What follows holds for $N = 2$ if we add to the Lagrangian a charge-two monopole operator. This breaks the $U(1)$ global symmetry down to $\mathbb{Z}_2$, which coincides with the magnetic symmetry for all other values of $N$.

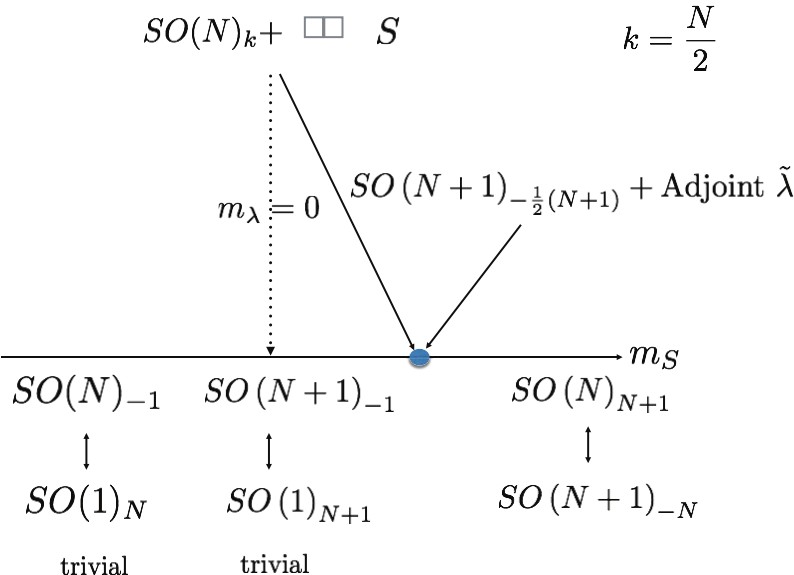

Figure 18: The phase diagram of $SO(N)_k$ gauge theory with a fermion in a symmetric-traceless tensor for $k = \frac{N}{2}$. There is one transition that connects a nontrivial TQFT and a trivial one. We propose a fermionic dual for the nontrivial transition.

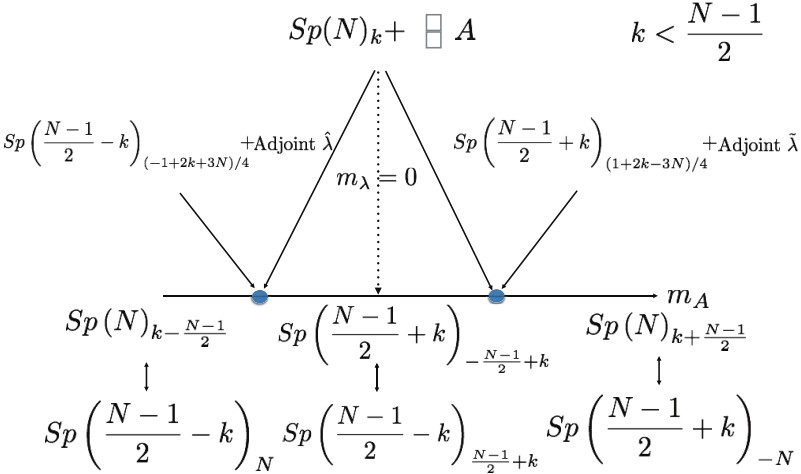

Figure 19: The phase diagram of $Sp(N)_k$ gauge theory with a fermion in an antisymmetric-traceless tensor for $k < \frac{N-1}{2}$. There are two phase transitions between different infrared TQFTs. We propose a dual fermionic description of the two transitions.

given by the first line of eq. (4.2). (It is easy to verify that for this range of $k$ the theories on the right hand side of eq. (4.2) are in the regime where they have only one transition according to the previous section. Hence, these dualities make sense.)

In $Sp(N)_k$ with a fermion $A$ and $k < T(R)/2 = (N-1)/2$ there is an intermediate phase, which is not visible semiclassically. This new phase is described by the nontrivial intermediate TQFT

$$Sp\left(\frac{N-1}{2} + k\right)_{-\frac{N-1}{2}+k} \longleftrightarrow Sp\left(\frac{N-1}{2} - k\right)_{\frac{N-1}{2}+k}. \tag{4.3}$$

The phase diagram for $Sp(N)_k$ for $k < (N-1)/2$ has two transitions, which admit dual

fermionic descriptions (see fig. 19). This suggests the fermion/fermion dualities

$$Sp(N)_k + \text{antisymmetric } A \longleftrightarrow Sp\left(\frac{N-1}{2}+k\right)_{-\frac{3N}{4}+\frac{k}{2}+\frac{1}{4}} + \text{adjoint } \tilde{\lambda} \,,$$

$$Sp(N)_k + \text{antisymmetric } A \longleftrightarrow Sp\left(\frac{N-1}{2}-k\right)_{\frac{3N}{4}+\frac{k}{2}-\frac{1}{4}} + \text{adjoint } \hat{\lambda} \,. \qquad (4.4)$$

And again, they are related.

The ultraviolet gauge theories with $k = 0$ at the point where the fermion is massless are time-reversal invariant, and the infrared TQFTs for $k = 0$ are $SO\left(\frac{N+2}{2}\right)_{\frac{N+2}{2}}$ and $Sp\left(\frac{N-1}{2}\right)_{\frac{N-1}{2}}$, which are indeed time-reversal invariant.

We would like to analyze now the time-reversal 't Hooft anomaly in these theories. We start with $Sp(N)_0$ with an antisymmetric-traceless fermion $A$. This theory exists only for odd $N$. In the ultraviolet the fermions contribute $\nu_{UV} = (N(2N-1)-1) \bmod 16 = (-N+1) \bmod 16$. In the infrared we have the TQFT $Sp\left(\frac{N-1}{2}\right)_{\frac{N-1}{2}}$. Using $\nu(Sp(n)_n) = \pm 2n \bmod 16$ we find, with an appropriate choice of sign, that the anomaly of the infrared theory is $\nu_{IR} = (-N+1) \bmod 16$, as in the ultraviolet.

Next, we move to the time-reversal 't Hooft anomaly in $SO(N)_0$ with a symmetric-traceless fermion $S$, which exists only for even $N$. For $N = 2 \bmod 4$, the anomaly in the ultraviolet is $\nu_{UV} = \left(\frac{N(N+1)}{2} - 1\right) \bmod 16 = \left(\frac{N}{2} + 1\right) \bmod 16$. In the infrared we have the TQFT $SO\left(\frac{N+2}{2}\right)_{\frac{N+2}{2}}$. Using $\nu(SO(n)_n) = \pm n \bmod 16$ we find, with an appropriate choice of sign, that the anomaly of the infrared theory is $\nu_{IR} = \left(\frac{N}{2} + 1\right) \bmod 16$, as in the ultraviolet.

As already noted in discussing adjoint QCD, the symmetry, whose anomalies should be matched for $SO(N)_0$ with $N = 0 \bmod 4$ is $CT$ rather than $T$ [58]. (In particular, in all the cases the symmetry that we discuss has an unambiguous, gauge invariant, anomaly.) For a symmetric-traceless fermion $S$ the $CT$ 't Hooft anomaly in the ultraviolet is $\nu_{UV} = \left(\frac{N(N-1)}{2} - (N-1)\right) \bmod 16 = \left(\frac{N+2}{2}\right) \bmod 16$. Using $\nu(SO(n)_n) = \pm n \bmod 16$ and that the infrared TQFT is $SO\left(\frac{N+2}{2}\right)_{\frac{N+2}{2}}$ we find, with an appropriate choice of sign, that the anomaly of the infrared theory is $\nu_{IR} = \left(\frac{N+2}{2}\right) \bmod 16$, as in the ultraviolet.

It is noteworthy that unlike the theories with an adjoint fermion, here the anomalies match without an added massless Majorana fermion. This is satisfying because the theories discussed in this section are not supersymmetric.

We consider the matching of the $T$ and $CT$ 't Hooft anomalies in these $SO$ and $Sp$ theories as nontrivial checks of our scenarios.

# Acknowledgments

We would like to thank M. Barkeshli, F. Benini, M. Cheng, C. Cordova, D. Gaiotto, P.-S. Hsin, A. Kapustin, M. Metlitski, S. Todadri, C. Vafa, and E. Witten for useful discussions. The work of NS was supported in part by DOE grant DE-SC0009988. J.G. would like to thank the Simons Center for warm hospitality. This research was supported in part by Perimeter Institute for Theoretical Physics. Research at Perimeter Institute is supported by the Government of Canada through Industry Canada and by the Province of Ontario through the Ministry of Research and Innovation. J.G. also acknowledges further support from an NSERC Discovery Grant and from an ERA grant by the Province of Ontario. Z.K. is supported in part by an Israel Science Foundation center for excellence grant and by the I-CORE program of the Planning and Budgeting Committee and the Israel Science Foundation (grant number 1937/12). Z.K. is also supported by the ERC STG grant 335182 and by the Simons Foundation grant 488657

(Simons Collaboration on the Non-Perturbative Bootstrap). Any opinions, findings, and conclusions or recommendations expressed in this material are those of the authors and do not necessarily reflect the views of the funding agencies.

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
