# Peer review of "Phases Of Adjoint QCD$_3$ And Dualities"

_SciPost Physics, doi:SciPost Phys. 5, 007 (2018)_

## Round 3 · Referee Report · Anonymous · 2018-2-20

Strengths
1. An analysis of 3d adjoint QCD as a function of Chern-Simons level and gauge group, using all the tools that have been put to work to study 3d bosonization dualities.
2. Clarity of presentation.
Weaknesses
1. None come to mind, really.
Report
The authors present a conjecture for the phase diagram of three-dimensional gauge theory with a Chern-Simons term coupled to an adjoint Majorana fermion. The authors use the same sort of logic that have been put to work to argue for non-supersymmetric ``3d bosonization'' dualities of late, exploiting level/rank duality in massive phases, matching global symmetries and their 't Hooft anomalies, and so on. In this setting, there is one more piece of data the authors can exploit, namely an $\mathcal{N}=1$ supersymmetry at a particular value of the fermion mass, which is unbroken at large Chern-Simons level and spontaneously broken at small level.
At large Chern-Simons level, all evidence is consistent with the existence of two massive, topologically non-trivial phases, mediated by some transition (which is second order at large N, but whose order is unknown at small N, k). If one has $SU(N)_k$ gauge theory, then the two phases are described by $SU(N)$ Chern-Simons theory with a one-loop shifted level as obtained from integrating out a massive adjoint fermion, $SU(N)_{k \pm \frac{N}{2}}$. If one regulates the theory by adding a small Yang-Mills term, then these phases are visible ``semiclassically,'' i.e when one takes the fermion mass to be much larger than the scale set by the Yang-Mills coupling and the theory is perturbative.
At moderate to low Chern-Simons level, following the same sort of logic used last summer in a paper by two of the authors in the context of 3d QCD, the authors present evidence that the two semiclassical phases are separated by two phase transitions with a a new ``quantum phase'' (which is gapped) in between, which for $SU(N)_k$ theory coupled to a fermion is a $U(N')$ theory, and near the two transitions, the theory can be described by a dual $U(N')$ theory coupled to an adjoint fermion. Crucially, in order for the story to hang together, there is no single dual $U(N')$ theory coupled to matter which describes both the transitions, and the massive phase in between. The authors perform a similar analysis for $SO$ and $Sp$ gauge theory, coupled to a Majorana fermion in the symmetric two-index and antisymmetric two-index representations.
The manuscript is clearly written, timely, and interesting, and I am nearly ready to recommend publication in SciPost. Before publication, I do have a couple of quick questions and comments. My questions are:
1. I was wondering if the authors could elaborate a bit on their Footnote 12. From the point of view of the dual descriptions near the transitions near A and B, I suppose I should no longer trust those descriptions once the fermion mass is comparable to the other scales in the problem, and that this should happen at some point in between the two transitions. Is that right?
2. Regarding the time-reversal anomaly $\nu$ for the $k=m_{\lambda}=0$ theory, there is some text below Eq. (2.9) which I find slightly confusing. It is not clear whether the authors have truly matched $\nu$ in this case, or if they have shown that it may match. Would the authors mind elaborating briefly on this point?
Finally, the comment. I was trying to think of some additional tests that may support the authors' conjecture. Two tests came to mind, but they ended up working out trivially. Sometimes trivial tests are not worth mentioning, and sometimes they are. I'll let the authors judge. The first is that, even though neither adjoint QCD nor the dual conjectured by the authors at small $k$ have baryons or monopoles, one may match ``baryon vertices'' of $SU(N)_k$ adjoint QCD with ``monopole vertices'' of the dual $U(N')$ description. But this matching seems to be insensitive to the fermion on both sides. The second was to match the spectrum of gauge-invariant operators at large $N$, $k$, but these are just fermion bilinears with derivatives.
Requested changes
See report for my questions.
I do have one comically anal change I'd recommend. Let the authors do what they will. In the various figures the authors present for phase diagrams, they use arrows to denote which gauge theories describe various critical points in the diagram. Truth be told, the arrows don't look like arrows; they look like lines. I don't want to say how long I puzzled over whether they were really critical lines, not arrows, and if there was really a second axis in the phase diagram. Anyway I'd recommend moving the head of the arrow slightly away from the critical points and make the head slightly bigger.
Jaume Gomis on 2018-02-26 [id 218]
We would like to first thank the referee for a thoughtful report and for a positive response!
We briefly answer here the questions:
1) It is not uncommon that the low energy description of a theory has multiple patches with different dual descriptions with limited radius of applicability. In our case the dual theory always is in the ``two-phase" region and indeed covers the transition where two phases coincide.
2) As we mention in the text, the sign of the time reversal anomaly is not determined, but we note (and this is true for all theories studied in the paper), that there is always a choice of sign such that the anomalies beautifully match! This is rather nontrivial indeed. We make this point several times in the paper and feel this should suffice.
Finally, we thank the referee for suggestion on figures. We believe that given the careful captions written for each figure that there should not be a confusion.
We like to thank the referee again for the comments!

---

## Round 3 · Referee Report · Anonymous · 2018-2-21

Strengths
Clear exposition of new interesting results.
Weaknesses
Somewhat confusing figures. (More will be explained in the report.)
Report
This is an extremely nice paper where the IR behavior of the 3d SU(N), SO(N), Sp(N) theories with adjoint fermions and at Chern-Simons level k is analyzed for all k and N. This is done by a masterly combination of the level-rank dualities, the anomaly matching of the time-reversal symmetry, etc. The referee does not find nothing major to be revised, but some minor points can be suggested, which will be given below.
1. Fig. 1 and Fig. 2 were confusing. The referee confesses that only after coming to, say, Fig 3, he understood what various lines meant. For example, when the referee first saw Fig. 1, he didn't understand what the vertical direction supposed to mean, why one arrow is dotted and why the other is not, whether it is meaningful that $m_\lambda=0$ is written around the middle of the figure, compared to $m_\lambda<0$ and $m_\lambda>0$ written at the bottom... The referee agrees that once he saw Fig.7 with all the other information, the meaning became apparent, but a slightly nicer presentation of some of the earlier figures would not hurt the reader.
2. It would be slightly nicer if the title of Sec. 2.2 is |k|<N/2 and that of Sec. 3.2 is |k|<h/2.
3. The authors repeatedly mention that two dualities describing two transition points are related by "analytic continuation and the orientation reversal." It would be nice if there is a clarification whether this is supposed to be just a mnemonic or this has something more physical in it.
4. In p.14, the authors refer to a $\mathbb{Z}_8$ classification of 3d topological terms for $\mathbb{Z}_2$ bundles on spin manifolds, and refer to three papers [52,53,54]. It would be instructive to the readers if the authors point out that this $\mathbb{Z}_8$ just reflects the standard 2d anomaly of chiral GSO projection.
Here is a short explanation (which is probably unnecessary): having a $\mathbb{Z}_2$ bundle and a spin structure is the same as having two spin structures, whose difference is the $\mathbb{Z}_2$ bundle. Therefore, the 3d spin $\mathbb{Z}_2$ SPTs correspond to the anomalies of 2d systems with two spin structures. Typical such systems are just 2d non-chiral majorana fermions, coupled to left-moving and right-moving spin structures independently. The anomaly arises when the left-moving and right-moving spin structures can't be summed over independently. For a single majorana fermion, the R-NS sector has spin 1/16. Therefore, only with eight copies, one has spin 1/2, which is the minimal allowed value for a spin theory.
(Type II strings can have a consistent chiral GSO projection only because the number of Majorana fermions in the light cone gauge is a multiple of 8. This is a less appreciated "numerical accident" supporting the consistency of the string theory.) This alone explains that the group of 3d spin $\mathbb{Z}_2$ SPT contains $\mathbb{Z}_8$; so, the $\mathbb{Z}_8$ classification was in essence known to string theorists for more than three decades. The new part, in the refere's understanding, is that this exhausts all the possible 3d spin $\mathbb{Z}_2$ SPT.
The referee understands that the Appendix B of [48] essentially contains this explanation, since in the equation (B.1) there, the $SO(L)_1[0]$ in the denominator can be thought of as the right-moving $L$ Majorana fermions with the right moving spin structure, and the $SO(L)_1[B]$ in the numerator can be thought of as the left-moving $L$ Majorana fermions with the left-moving spin structure which is different from the right-moving one by $B$.
That said, the paper [48] would have been slightly more instructive too, if it contains a short mention on GSO projections, just for the sake of the readers.
Somehow this SPT viewpoint on the GSO projection is never explicitly mentioned on the hep-th side of the community, to the knowledge of the referee. It is sometimes mentioned on the cond-mat side, see e.g. a paper by Ryu et al. https://arxiv.org/abs/1202.4484 , although the referee is not sure if that paper by Ryu is the earliest one which pointed out the connection (or equivalence) to the GSO projection.
Since one of the authors of this manuscript under review is also one of the authors of the influential paper "Spin Structures in String Theory" with E. Witten, the referee just wanted to bring this point into the attention of the authors.
5. In Sec. 2.3 and other sections where the T anomaly is studied, the authors assumes that the UV anomaly simply comes from the fermions. Is it clear that the gauge field does not contribute? As the anomaly of discrete symmetries are subtler than the more traditional anomalies captured in anomaly polynomials, it would be nicer if there is at least a sentence explaining that it does not contribute.
6. Above (3.5), "this suggest" should be "this suggests."
7. In (3.3), the authors discuss CT vs. T. Does the fact that the anomaly of naive T is gauge dependent mean that there is a mixed anomaly between T and SO(N)? If so, can it be used for 't Hooft anomaly matching?
8. Around the middle of p.31, "Intuitive," should be "Intuitively."
Requested changes
As all the changes are just suggested rather than requested, they are written in the report section.
Author: Jaume Gomis on 2018-03-27 [id 234]
(in reply to Report 2 on 2018-02-21)
We would like to thank the referee for the very thoughtful report, for catching typos and for a very insightful comment.
We have corrected the typos found by the referee, added small clarifications to the captions to the figures in introduction to help with the figures and added a footnote connecting with the GSO projection as suggested by the referee.
Thanks again

---

## Round 3 · Referee Report · Anonymous · 2018-2-24

Strengths
A very clear and detailed discussion of the complicated phase diagram that arises from a simple quantum field theory. It is exciting that the techniques developed in recent years finally allow us to map out low energy physics of systems like this.
Weaknesses
The authors evident skill with physics is not matched by their skill in drawing clean, readable diagrams.
Report
This is an excellent paper, exploring the phase diagram of a simple d=2+1 quantum field theory: SU(N) Yang-Mills, with Chern-Simons level k, coupled to an adjoint Majorana fermion. (Generalisations to SO(N) and Sp(N) are also presented.)
For suitably large k, is is straightforward to guess the phase diagram: for large fermion mass, the theory flows to two different gapped, topological phases depending on whether the mass is positive or negative. These are then connected by a single phase transition as the mass is varied.
For small k, things are much more subtle. The authors argue convincingly that the theory admits a hidden "quantum phase" at small masses, so you can reach one semi-classical region from the other only by going via this third phase. The need for this third phase follows from an analysis of the supersymmetric point (where it has long been known that susy is broken) coupled with matching the 1-form anomalies.
Having identified this third, intermediate phase, the authors then use this knowledge to offer descriptions of the phase transitions. This leads to yet more entries in the ever-growing list of dualities among non-supersymmetric 3d gauge theories. Compelling evidence for these dualities is presented.
I have no hesitation at all in recommending this paper for publication.
Requested changes
None
Author: Jaume Gomis on 2018-03-27 [id 233]
(in reply to Report 3 on 2018-02-24)We would like to thank the referee for a nice report. We have added some clarifications to the captions of the figures in the introduction to help reading the figures

---

## Round 3 · Referee Report · Anonymous · 2018-3-23

Strengths
What I said before.
Weaknesses
None come to mind, really
Report
I'd like to thank the authors for their reply. I have an extremely minor comment to make before going on to publication. I suppose I was a little too indirect in my original comments about the time-reversal anomaly in my first report. The minor complaint I had in mind was that, in the Introduction, the authors claim to compute the time-reversal anomaly $\nu$ in various theories. But then what they do in practice is something different, namely to propose the T-phase for the emergent low-energy Chern-Simons gauge theory, in such a way that $\nu$ is matched.
This is of course a non-trivial and convincing check on the authors proposal. But I would quibble with this internal inconsistency.
All of that being said, this is an O($\epsilon$) or perhaps even an O($\epsilon^2$) issue and I don't think it is sufficiently important as to delay publication in SciPost.
Requested changes
See report.

---

## Editorial Decision

published